# Amplification of heat extremes by plant $CO_2$ physiological forcing

Christopher B. Skinner[1], Christopher J. Poulsen [iD] [1] & Justin S. Mankin [iD] [2,3,4]

Plants influence extreme heat events by regulating land-atmosphere water and energy exchanges. The contribution of plants to changes in future heat extremes will depend on the responses of vegetation growth and physiology to the direct and indirect effects of elevated $CO_2$. Here we use a suite of earth system models to disentangle the radiative versus vegetation effects of elevated $CO_2$ on heat wave characteristics. Vegetation responses to a quadrupling of $CO_2$ increase summer heat wave occurrence by 20 days or more—30–50% of the radiative response alone—across tropical and mid-to-high latitude forests. These increases are caused by $CO_2$ physiological forcing, which diminishes transpiration and its associated cooling effect, and reduces clouds and precipitation. In contrast to recent suggestions, our results indicate $CO_2$-driven vegetation changes enhance future heat wave frequency and intensity in most vegetated regions despite transpiration-driven soil moisture savings and increases in aboveground biomass from $CO_2$ fertilization.

[1] Department of Earth and Environmental Sciences, University of Michigan, 1100 North University Ave, Ann Arbor, MI 48104, USA. [2] Ocean and Climate Physics, Lamont-Doherty Earth Observatory, 61 Route 9W, P.O Box 1000, Palisades, NY 10964, USA. [3] NASA Goddard Institute for Space Studies, New York, NY 10025, USA. [4] Department of Geography, Dartmouth College, Hanover, NH 03755, USA. Correspondence and requests for materials should be addressed to C.B.S. (email: chrisbs@umich.edu)

Prolonged exposure to extreme heat, such as during a heat wave, imposes severe stresses on natural and human systems. Acute heat-related impacts include increased human morbidity and mortality[1,2], loss of livestock and crop failure[3], increased wildfires[4], and reductions in vegetation gross primary productivity (GPP)[5]. Heat wave frequency has recently increased over many parts of the globe, highlighting the exceptional sensitivity of heat extremes to even small changes in mean global warming[6,7]. Given continued anthropogenic emissions of $CO_2$, climate models project further increases in heat wave occurrence, intensity, and duration, spurring a considerable effort to diagnose the processes that shape these events in current and future climates[8–10]. Recent work has called attention to the potentially substantial, but largely unexplored and uncertain impact of $CO_2$-vegetation interactions, including $CO_2$ physiological forcing and $CO_2$ fertilization, on future heat wave characteristics[11,12]. Here, we analyze a suite of Earth system models (ESMs) from the Coupled Model Intercomparison Project Phase 5 (CMIP5)[13] in a set of idealized experiments to quantify the contribution of the vegetation response to $CO_2$ to projected heat wave changes and to better understand its role in shaping regional-scale and inter-model differences in projected future heat extremes.

In addition to serving as a greenhouse gas, $CO_2$ indirectly influences climate through its impact on vegetation growth and physiology. Under high $CO_2$, plant photosynthetic carbon fixation rates increase, while stomatal aperture is reduced or maintained[14]. The increase in photosynthesis and subsequent enhanced biomass production due to the non-radiative effects of higher $CO_2$ is known as $CO_2$ fertilization[15,16]. The closing of stomata and subsequent reduction in stomatal conductance and transpiration (evaporation of water from the leaf interior) due to the non-radiative effects of higher $CO_2$ is known as $CO_2$ physiological forcing[17,18]. While the magnitudes of the fertilization and physiological responses vary by plant species and under different environmental conditions, including water, light, and nutrient availability, observational and modeling evidence suggests that the two effects have opposing influences on climate[19,20]. Though enhanced $CO_2$ often has a limited impact on leaf area index (LAI, defined as one-sided leaf area per unit ground surface area), in nutrient-limited regions, and in mature forests[21], $CO_2$ fertilization can lead to enhanced LAI during the early stages of plant development[21,22] and in regions that are water-limited[23]. Greater LAI can enhance plant transpiration and surface evaporative cooling given sufficient moisture supply[24,25]. Meanwhile, in regions that are not severely water-limited, $CO_2$ physiological forcing limits transpiration and enhances the ratio of sensible to latent heat fluxes at the leaf surface, increasing boundary layer temperatures[17,26,27].

Future changes in evapotranspiration (ET, the sum of transpiration and soil and canopy evaporation) have the potential to alter the characteristics of extreme heat events, such as heat waves. Heat waves are initiated when large-scale anticyclonic atmospheric patterns become stagnant[7,28]. The severity of heat waves is amplified through feedbacks with the underlying land surface. Dry soils and low ET enhance surface sensible heating of the atmosphere and promote higher near-surface temperatures during the heat wave[29].

Given the impact of $CO_2$ physiological forcing and $CO_2$ fertilization (hereafter collectively referred to as $CO_2$ vegetation forcing) on surface moisture and energy fluxes, each may contribute to projected changes in future heat wave events, but the net effects on projected heat waves remain unresolved. The vegetation response to elevated $CO_2$ could mitigate the frequency and intensity of summer heat waves by increasing the springtime canopy water use efficiency (WUE) via $CO_2$ physiological forcing, thus increasing the soil water available for evapotranspirative cooling later in the peak of summer[12,30,31]. For example, observed growing season water savings from $CO_2$-induced transpiration reductions range from up to ~2.15 mm day$^{-1}$ in some grassland ecosystems (exposed to 100% increase in $CO_2$)[30], to ~0.6 mm day$^{-1}$ in some temperate deciduous tree species (exposed to 40% increase in $CO_2$)[22]. Coupled with increasing summertime LAI via $CO_2$ fertilization, regions with greater soil moisture, particularly those that would have otherwise become water-limited during the summer, could see increases in transpiration, and, therefore, surface cooling when transpiration demand is high (such as during dry, hot days)[12]. Indeed, observational and modeling work shows that irrigated cropland has experienced fewer summer hot extremes in recent decades as a result of enhanced ET from greater soil moisture[32,33]. On the other hand, reduced stomatal conductance from $CO_2$ physiological forcing during the summer could diminish transpiration and the associated evaporative cooling in spite of increased LAI and soil moisture, elevating summer temperatures[34,35], allowing heat waves to increase in frequency, intensity, and length.

Most analyses of future heat extremes come from climate model simulations in which only the $CO_2$ radiative forcing is included, or from simulations in which $CO_2$ radiative forcing, $CO_2$ physiological forcing, and $CO_2$ fertilization are included simultaneously (e.g., ref.[36]). Furthermore, previous work on $CO_2$ vegetation forcing focuses primarily on the mean temperature response, whether at seasonal or annual time scales, leaving the possibility that projected future $CO_2$ vegetation forcing influences the warm extremes of the daily temperature distribution differently than the mean of the distribution. Here we analyze a suite of ESMs from CMIP5 that include active biogeophysics and biogeochemistry to study the impact of $CO_2$ vegetation forcing (via a quadrupling of $CO_2$ concentrations) on four indices of heat extremes. Our results indicate that the vegetation response to elevated $CO_2$, primarily through reduced stomatal conductance and the hydrologic responses it induces, exacerbates $CO_2$ radiative-driven increases in extreme heat frequency and intensity.

## Results

**LAI and transpiration responses to $CO_2$ vegetation forcing.** Elevated $CO_2$ concentrations increase mean summer LAI via $CO_2$ vegetation forcing (see Methods and Tables 1–2) in each of the six CMIP5 models (see Supplementary Table 1) across all latitudes (JJAS in the Northern Hemisphere, DJFM in the Southern Hemisphere) (Fig. 1a, c, e, g). The stimulation of LAI by increasing $CO_2$ ($\Delta$LAI/$\Delta CO_2$ ppm) is greatest at relatively low $CO_2$ concentrations (Supplementary Table 2). LAI stimulation diminishes at higher $CO_2$ levels, indicating a trend towards LAI

**Table 1 CMIP5 simulation names and associated $CO_2$ forcing**

| Simulation name | $CO_2$ radiative forcing | $CO_2$ physiological forcing | $CO_2$ fertilization |
|---|---|---|---|
| TotalCO2 (1pctCO2) | Yes $CO_2$ increases from 284 to 1132 ppm | Yes $CO_2$ increases from 284 to 1132 ppm | Yes $CO_2$ increases from 284 to 1132 ppm |
| RadCO2 (esmFdbk1) | Yes $CO_2$ increases from 284 to 1132 ppm | No $CO_2$ fixed at 284 ppm | No $CO_2$ fixed at 284 ppm |
| VegCO2 (esmFixClim1) | No $CO_2$ fixed at 284 ppm | Yes $CO_2$ increases from 284 to 1132 ppm | Yes $CO_2$ increases from 284 to 1132 ppm |

**Table 2 CO$_2$ forcing experiments**

| CO$_2$ forcing type | Calculated as difference between | Reference climate for heat wave definition | Displayed in figures |
|---|---|---|---|
| CO$_2$ vegetation forcing (avg CO$_2$ 984 ppm) | Years 111–140 of *TotalCO2* minus Years 111–140 of *RadCO2* | Years 111–140 of *RadCO2* | Figures: 1,2,3,4,5; Supp. Figures: 1,2,3,4,5,6,7,9 |
| CO$_2$ radiative forcing (avg CO$_2$ 984 ppm) | Years 111–140 of *TotalCO2* minus Years 111–140 of *VegCO2* | Years 111–140 of *VegCO2* | Figures: 4; Supp. Figures: 6,7 |
| CO$_2$ vegetation forcing (avg CO$_2$ 575 ppm) | Years 58–87 of *TotalCO2* minus Years 58–87 of *RadCO2* | Years 58–87 of *RadCO2* | Supp. Figure: 8 |
| CO$_2$ vegetation forcing (avg CO$_2$ 984 ppm) | Years 111–140 of *VegCO2* minus Years 1–29 of *VegCO2* | Years 1–29 of *VegCO2* | Supp. Figure: 8 |
| CO$_2$ radiative forcing (avg CO$_2$ 984 ppm) | Years 111–140 of *RadCO2* minus Years 1–29 of *RadCO2* | Years 1–29 of *RadCO2* | Supp. Figure: 8 |
| Total CO$_2$ forcing (avg CO$_2$ 984 ppm) | Years 111–140 of *TotalCO2* minus Years 1–29 of *TotalCO2* | Years 1–29 of *TotalCO2* | Figure: 4 |

saturation throughout the tropics, subtropics, and extratropics at the highest CO$_2$ concentrations (Fig. 1a, c, e). The largest changes in the magnitude of LAI from end-of-21$^{st}$-century CO$_2$ vegetation forcing (average CO$_2$ ~984 ppm) are located in forested regions of the tropics and mid-latitudes where summer season LAI is high in the reference (baseline) climate period (Fig. 1g and Supplementary Fig. 2a-l). An evaluation of LAI in each model is provided in the Supplementary Information (Supplementary Note 1 and Supplementary Fig. 1) and in ref.[37].

Despite widespread LAI enhancement, average summer transpiration goes down in response to CO$_2$ vegetation forcing (Fig. 1b, d, f, h). In the tropics, the average rate of transpiration change over the course of CO$_2$ doubling ($\Delta$Tran/$\Delta$CO$_2$ ppm) levels off very slightly from the first CO$_2$ doubling (~284 ppm to 568 ppm) to the second CO$_2$ doubling (~568 ppm to 1132 ppm) in most models (note, ET is used for HadGEM2-ES) (Fig. 1b and Supplementary Table 2). In the subtropics and extratropics, CO$_2$ vegetation forcing initially enhances mean summer transpiration in a few models as increases in LAI potentially outpace reductions in stomatal conductance (though other climate system processes such as changes in rainfall may also contribute to the change) (Fig. 1d, f and Supplementary Table 2). However, most models exhibit reductions in transpiration throughout the 140-year simulations, and all models exhibit transpiration reductions over the course of the second doubling of CO$_2$ in the tropics, subtropics, and extratropics (Fig. 1b, d, f, and Supplementary Table 2). Transpiration reductions are concentrated in forested regions of the tropics and the Northern Hemisphere mid-to-high latitudes (Fig. 1h). Most models exhibit statistically significant reductions in transpiration in tropical Africa, Southeast Asia, and tropical South America, as well as throughout forested regions of mid-latitude North America, Europe, and Asia (Fig. 1h and Supplementary Fig. 2s-x). These regions correspond to the locations of climatologically high summer canopy density and transpiration in the reference climate period (Supplementary Fig. 2a-f and 2m-r). The results highlight the role of reduced stomatal conductance in shaping projected transpiration change.

CO$_2$ vegetation forcing simultaneously increases summer LAI and reduces summer transpiration over most of Earth's land surface (Fig. 2a). Across the models, between 58.4% (BCC-CSM1-1) and 86.7% (CESM1-BGC) of land area exhibits both positive LAI changes and negative transpiration changes (Fig. 2a). Increases in summer season transpiration due to CO$_2$ vegetation forcing are primarily confined to grassland and steppe regions in semi-arid portions of North America, Asia, Africa, and Australia, covering between 9.4% (CESM1-BGC) and 37.62% (BCC-CSM1-1) of total land area in the models (Fig. 2a and Supplementary Figure 2s-x). These regions of increasing summer transpiration exhibit relatively low summer LAI in the reference climate

simulations (Supplementary Fig. 2a-f) coupled with large percentage increases in LAI in response to CO$_2$ vegetation forcing (Supplementary Fig. 3a-f). This reveals that, in general, unless LAI increases substantially, a vegetated grid cell will exhibit reduced transpiration in response to future CO$_2$ vegetation forcing from high CO$_2$. Summer season WUE, defined as the ratio of summer GPP to summer transpiration, increases across all biomes (Supplementary Fig. 3g-l). The percent change in WUE is particularly large in the relatively warm semi-arid regions that exhibit mean increases in transpiration in response to CO$_2$ vegetation forcing (Supplementary Fig. 3g-l). This result highlights the important point that increases in WUE do not necessarily imply a reduction in total plant water use if photosynthesis and biomass increase[38].

Though most ESMs underestimate the global ratio of transpiration to total ET (Fig. 2b)[39], the reductions in transpiration from CO$_2$ vegetation forcing in the models, which, globally-averaged range from −7.5 mm summer$^{-1}$ in BCC-CSM1-1 to −29.5 mm summer$^{-1}$ in CESM1-BGC, are large enough to drive statistically significant summer ET declines in most forested regions (Supplementary Fig. 4a-f). As indicated by the decrease in evaporative fraction (the ratio of latent heat fluxes to the sum of latent and sensible heat fluxes), sensible heat fluxes increase at the expense of latent heat fluxes (and thus evaporative cooling) throughout the tropics and in regions of dense tree cover in the mid and high latitudes (Fig. 3a and Supplementary Fig. 4g-l). Enhanced boundary layer turbulence from greater sensible heat fluxes combined with reduced moisture fluxes from the surface reduces cloud formation and rainfall in many mid-to-high latitude locations, enhancing surface solar insolation (Supplementary Fig. 4m-x). In general, the greatest summer near-surface warming (daily maximum temperature) occurs in the mid and high latitudes where the evaporative fraction, cloud cover, and rainfall decrease (Fig. 3a, and Supplementary Figs. 4g-x and 5a-f). Increases in mean daily maximum tropical temperatures are slightly smaller despite large reductions in transpiration because cloud cover and rainfall change very little or even increase in some areas in response to CO$_2$ vegetation forcing (Supplementary Figs. 4m-x and 5a-f)[35].

**Vegetation-driven temperature and heat extreme changes.** Mean maximum summer daily temperatures increase by more than 2 °C in response to CO$_2$ vegetation forcing throughout much of the Northern Hemisphere mid-to-high latitudes (Supplementary Fig. 5a-f). In tropical forests, most models project between 1 °C and 2 °C of warming (Supplementary Fig. 5a-f). With the exception of small areas in the subtropics and East Africa in CanESM2 and MPI-ESM-LR and a portion of the western

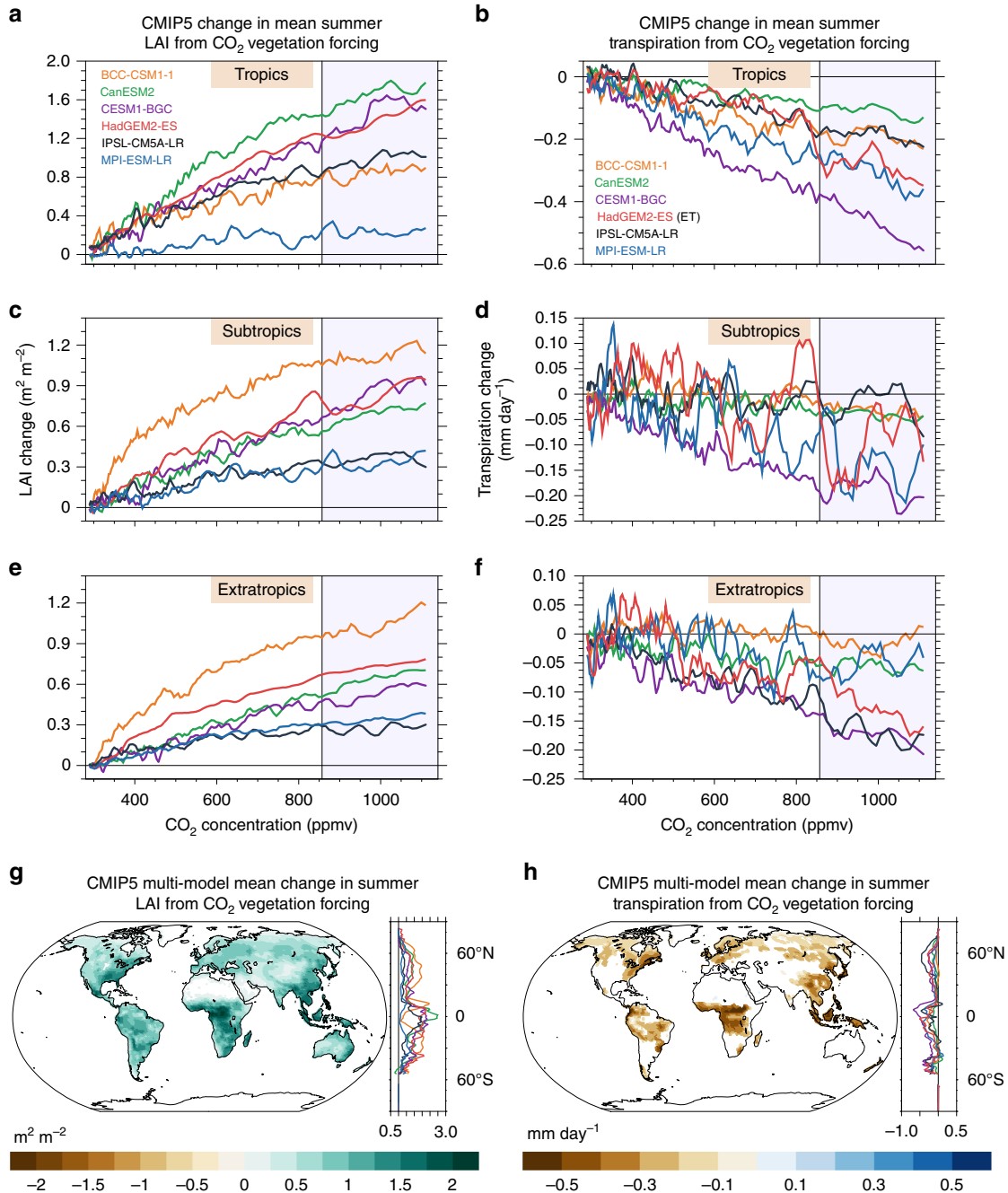

**Fig. 1** Change in leaf area index and transpiration from $CO_2$ vegetation forcing. Change in mean summer **a**, **c**, **e** leaf area index (LAI) and **b**, **d**, **f** transpiration from $CO_2$ vegetation forcing. For each $CO_2$ concentration, the area-weighted value for the **a–b** tropics (15°S–15°N), **c–d** subtropics (15°S/N–30°S/N), and **e–f** extratropics (30°S/N–70°S/N) is calculated for the *TotalCO2* and *RadCO2* simulations. Then, the values are smoothed with a five-year running mean and the difference (*TotalCO2 – RadCO2*) for each model is plotted. Multi-model mean change in summer **g** LAI and **h** transpiration from $CO_2$ vegetation forcing averaged over the final 30 years of the simulations (shaded region in **a–f**). Only statistically significant changes are plotted (Methods). Zonal averages of LAI and transpiration change (land grid points) for each model are plotted to right of each map. June–September (December–March) values are used in the Northern (Southern) Hemisphere. Evapotranspiration (ET) is used for HadGEM2-ES in **b**, **d**, **f**

Amazon in IPSL-CM5A-LR, no models exhibit statistically significant summer cooling due to $CO_2$ vegetation forcing (Supplementary Fig. 5a-f). Given that transpiration occurs primarily during daylight hours, $CO_2$ physiological forcing acts preferentially on daily maximum temperature, increasing the diurnal temperature range over vegetated regions (Supplementary Fig. 5g-l), though it should be noted that models do not accurately simulate nocturnal stomatal conductance[40].

$CO_2$ vegetation forcing also enhances the intensity of the hottest summer season days (Fig. 3b). The hottest daily maximum temperature experienced during the summer increases by ~2.5 °C across the Northern Hemisphere high latitudes, and by up to 2 °C across portions of the Amazon, Congo, and Southeast Asia. In some models, for example, CESM1-BGC and HadGEM2-ES, warming of the maximum daily summer temperature exceeds 3 °C over broad stretches of the tropics and the Northern Hemisphere mid and high latitudes (Supplementary Fig. 5m-r).

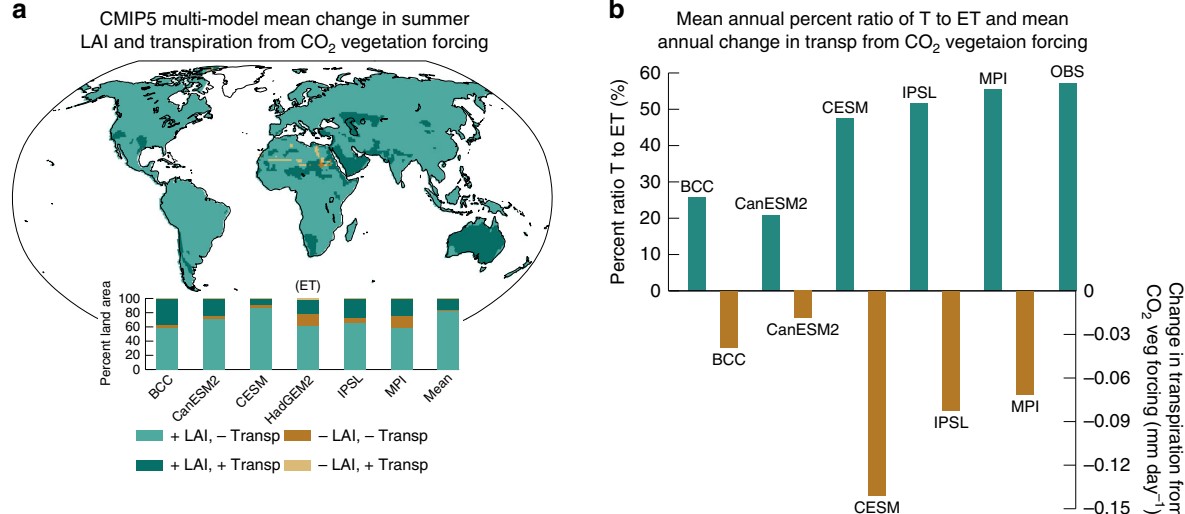

**Fig. 2** Changes in leaf area index and transpiration from $CO_2$ vegetation forcing. **a** Map of summer leaf area index (LAI) and transpiration changes of a particular direction for the CMIP5 multi-model mean. Bar plots of the percentage of land area with LAI and transpiration changes of a particular direction. **b** (top) Global mean annual percent ratio of transpiration to evapotranspiration (T to ET) from the CMIP5 models and from observations provided in Wei et al. (2017). **b** (bottom) Global mean annual change in transpiration in the CMIP5 models from $CO_2$ vegetation forcing. The values for the ratio of T to ET come from years 18–34 ($CO_2$ ~ 340–398 ppm) in each model's *TotalCO2* simulation. Years 18–34 are chosen to closely match the $CO_2$ values during the years 1982–2014, which are used in Wei et al. (2017). Area-weighted averages are calculated from 60°S–90°N and do not include Greenland. June–September (December–March) values are used in the Northern (Southern) Hemisphere. Evapotranspiration (ET) is used for HadGEM2-ES

The boundary layer and surface water and energy flux changes from $CO_2$ vegetation forcing alter heat wave characteristics across most vegetated areas of the globe. Models exhibit robust increases in total heat wave days (HWTD) each summer (Fig. 4a and Supplementary Fig. 6a-f). Similar to the pattern of mean warming, heat wave days increase most over forested areas of the mid-to-high latitudes and tropics. On average, large portions of North America, Europe, Asia, and the La Plata basin of South America experience between 15 and 25 more heat wave days each summer from $CO_2$ vegetation forcing. In the tropics, warming from $CO_2$ vegetation forcing leads to increases in excess of 30 heat wave days per summer season. In comparison to the pattern of heat wave day changes from $CO_2$ radiative forcing (see Methods, and Supplementary Fig. 6a-l), $CO_2$ vegetation forcing primarily impacts wetter, vegetated areas of the mid and high latitudes (Fig. 4a). In the Northern Hemisphere mid-to-high latitudes, heat wave day increases from $CO_2$ vegetation forcing are roughly 30 to 50% of the $CO_2$ radiative-driven response (Fig. 4e).

Note that the zonal average change in heat wave days from the total $CO_2$ forcing (see Methods) is plotted for illustrative purposes only (Fig. 4a). Given the choice of different reference climate for the heat wave definitions (see Methods), and because the radiative and vegetation forcings are not independent, the individual changes in heat wave characteristics from $CO_2$ vegetation forcing and $CO_2$ radiative forcing are not expected to sum to the total $CO_2$ response (Fig. 4a–d). For example, elevated $CO_2$ radiative forcing alone may be great enough to result in 100 additional summer heat wave days (out of 120 total summer days) in a particular location, while $CO_2$ vegetation forcing alone may cause sufficient warming to result in 40 more summer heat wave days in that same location.

The increases in summer heat wave days from $CO_2$ vegetation forcing are driven by more frequent and longer heat wave events (Fig. 4b, c, and Supplementary Figs. 6m-r and 7a-f). $CO_2$ vegetation forcing yields two to three more heat wave events per summer over the eastern US and eastern Europe, and over most of Canada and Asia. Tropical regions, including northwest South

America, equatorial Africa, and the Maritime Continent experience three to five more heat wave events per summer. While significant and notable, the vegetation effect on heat wave days and length is smaller than the effect of $CO_2$ radiative forcing alone (Supplementary Figs. 6g-l and 7g-l). The radiative response alone, for example, is sufficient to shift tropical and subtropical temperatures so far outside the range of natural variability that nearly all days in a climate with elevated $CO_2$ radiative forcing meet the heat wave day criteria, leading to long heat waves and fewer individual summer heat wave events in the tropics and subtropics in some models (Supplementary Fig. 6s-x).

$CO_2$ vegetation forcing increases the average length of the maximum summer heat wave event by more than eight days over the Amazon, Congo Basin, and Maritime Continent, and by as many as four to six days over much of the Northern Hemisphere mid-to-high latitudes (Fig. 4c). Heat wave intensity increases between 1 °C and 2 °C over portions of Canada, Europe, and Russia (Fig. 4d and Supplementary Fig. 7m-r). Smaller increases of 0.25 °C–1.5 °C are located in northwest South America, equatorial Africa, and the Maritime Continent. In several models, the increase in mid-to-high latitude heat wave intensity from $CO_2$ vegetation forcing is of similar magnitude to that from $CO_2$ radiative forcing (Fig. 4f and Supplementary Fig. 7m-x).

The simulated changes in heat wave metrics from $CO_2$ vegetation and CO2 radiative forcing are robust to the choice of the reference climate (see Methods). The impacts of elevated end-of-21$^{st}$-century $CO_2$ vegetation forcing and $CO_2$ radiative forcing within a historical period reference climate (relatively low $CO_2$) are very similar to those within a future reference climate (relatively high $CO_2$, see Supplementary Fig. 8). However, small differences in the changes in heat wave metrics from $CO_2$ vegetation forcing are present at high latitudes, where the choice of reference climate (low versus high $CO_2$) likely has a large impact on summer vegetation growth (compare Fig. 4a–d with Supplementary Fig. 8a-d). Not surprisingly, changes in heat wave metrics from projected mid-21$^{st}$ century $CO_2$ vegetation forcing (average $CO_2$ ~575 ppm, see Supplementary Fig. 8i-l) are smaller than those from the projected end of 21$^{st}$ century forcing (average

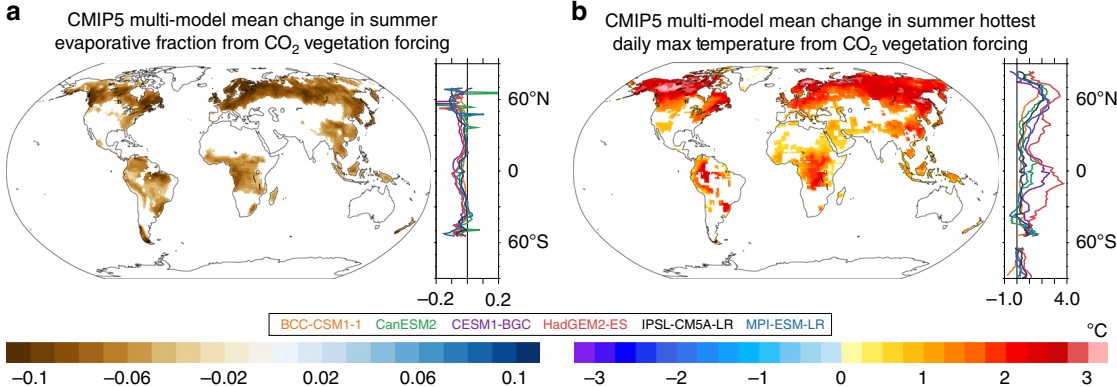

**Fig. 3** Change in the evaporative fraction and surface temperature from $CO_2$ vegetation forcing. Multi-model mean change in summer **a** mean evaporative fraction (EF) (ratio of latent heat fluxes to the sum of sensible and latent heat fluxes), and **b** mean hottest daily maximum 2-meter temperature from $CO_2$ vegetation forcing. Only statistically significant changes are plotted (Methods). Zonal averages of EF change and hottest daily maximum temperature change (land grid points) for each model are plotted to the right of each map. Note, the magnitudes of EF change north of 65°N and over Antarctica are very large in some models, and are masked out in the zonal average plot. June–September (December–March) values are used in the Northern (Southern) Hemisphere. Plots represent 30-year averages

$CO_2$ ~984 ppm, see Fig. 4a–d), though the spatial patterns of extreme heat change (where statistically significant) are similar. Overall, the six CMIP5 models analyzed in this study suggest that projected $CO_2$ vegetation forcing will increase both the mean and upper tail (heat extremes) of the summer daily temperature distribution in most vegetated regions of the tropics and mid-to-high latitudes, enhancing the risk of acute impacts associated with heat wave events.

**Growing season influence on summer season hydrology**. To assess whether hydrologic and vegetation changes outside of the summer season shape the vegetation and climate response to elevated $CO_2$ during the summer, we explored the temporal relationship between surface hydrology, LAI, and heat waves in four regions with robust summer season heat wave changes: northeastern North America, Europe, tropical South America, and tropical Africa (Fig. 5 and Supplementary Fig. 9). In the mid-latitude locations, reduced springtime transpiration from $CO_2$ vegetation forcing contributes to slightly greater total-column soil moisture at the start of the summer season in most models (Fig. 5a, b and Supplementary Fig. 9a–l, see Supplementary Table 1 for the hydrologically active soil column depths in each model). However, reductions in spring precipitation from $CO_2$ vegetation forcing (with the exception of BCC-CSM1-1 in North America) limit the accumulation of soil moisture entering the summer season. Though this excess soil moisture is available to some plants during the summer, it does not increase summer transpiration. Rather, the transpiration response to $CO_2$ vegetation forcing remains negative during the summer months, leading to further increases in soil moisture. Soil moisture accumulation during the summer is limited by reduced summer precipitation. The reduction in summer precipitation from $CO_2$ vegetation forcing, which manifests in fewer summer rainfall days[35], enhances the likelihood of conditions favorable for heat wave development. Changes in surface evaporation (canopy plus soil) are small and contribute little to the surface water and heat flux changes from $CO_2$ vegetation forcing during the year.

In the evergreen tropics, $CO_2$ fertilization and $CO_2$ physiological forcing impact climate throughout the year (Fig. 5c, d and Supplementary Fig. 9m–x). Similar to the mid-latitude locations, $CO_2$ vegetation forcing reduces austral summer transpiration in the Congo and Amazon basins regardless of elevated summer LAI and greater soil moisture at the start of the summer season. It is worth nothing that the small change in austral summer

transpiration and the large decrease in soil moisture in the Amazon in CanESM2 are the result of a substantial negative LAI bias (Supplementary Fig. 1), which diminishes the negative transpiration response in the multi-model mean plot (Fig. 5c). Additionally, the lack of austral summer transpiration decline in the Amazon in IPSL-CM5A is driven by a near 100% increase in LAI, a response not found in the other models (Supplementary Figs. 3e and 9q). Interestingly, most models do not exhibit a reduction in Amazon and Congo basin transpiration during the austral winter dry season (JJAS), when temperatures are at their climatological peak (Supplementary Figs. 9m-x). At the start of the dry season most models exhibit excess soil moisture from $CO_2$ vegetation forcing as a result of reduced austral spring-to-fall transpiration and, in some models, by increased summer-to-fall precipitation. The combination of excess soil moisture at the start of the dry season, greater LAI during the dry season, deep roots, and high evaporative demand are likely responsible for maintaining the same transpiration during this time (Fig. 5c) (e.g., ref. [12]). However, in most regions and times of the year, transpiration reductions in response to future $CO_2$ vegetation forcing in CMIP5 ESMs directly and indirectly (climate system feedbacks) lead to warmer temperatures and increased heat extremes regardless of soil moisture savings in previous months.

**Discussion**
Elevated $CO_2$ concentrations are expected to drive widespread increases in extreme heat events this century[41]. Most often, projections of future heat wave changes are attributed to the radiative impacts of higher $CO_2$. Here, we find that even without consideration of the radiative effects of $CO_2$, heat wave frequency will increase in vegetated regions as a consequence of vegetation's direct response to rising atmospheric $CO_2$ concentrations. Despite greater LAI from higher $CO_2$, reductions in stomatal conductance from $CO_2$ physiological forcing reduce warm season transpiration, limiting surface evaporative cooling, thus shifting both summer mean and extreme temperatures upward.

The greatest reductions in summer transpiration and subsequent increases in summer heat wave frequency, duration, and magnitude are located in tropical and mid-to-high latitude regions with dense tree cover and high climatological transpiration (Figs 1h, 4). These are regions in which transpiration is not strongly limited by water availability during much of the summer. The only areas where summer transpiration consistently increases in response to the combined influences of $CO_2$ physiological

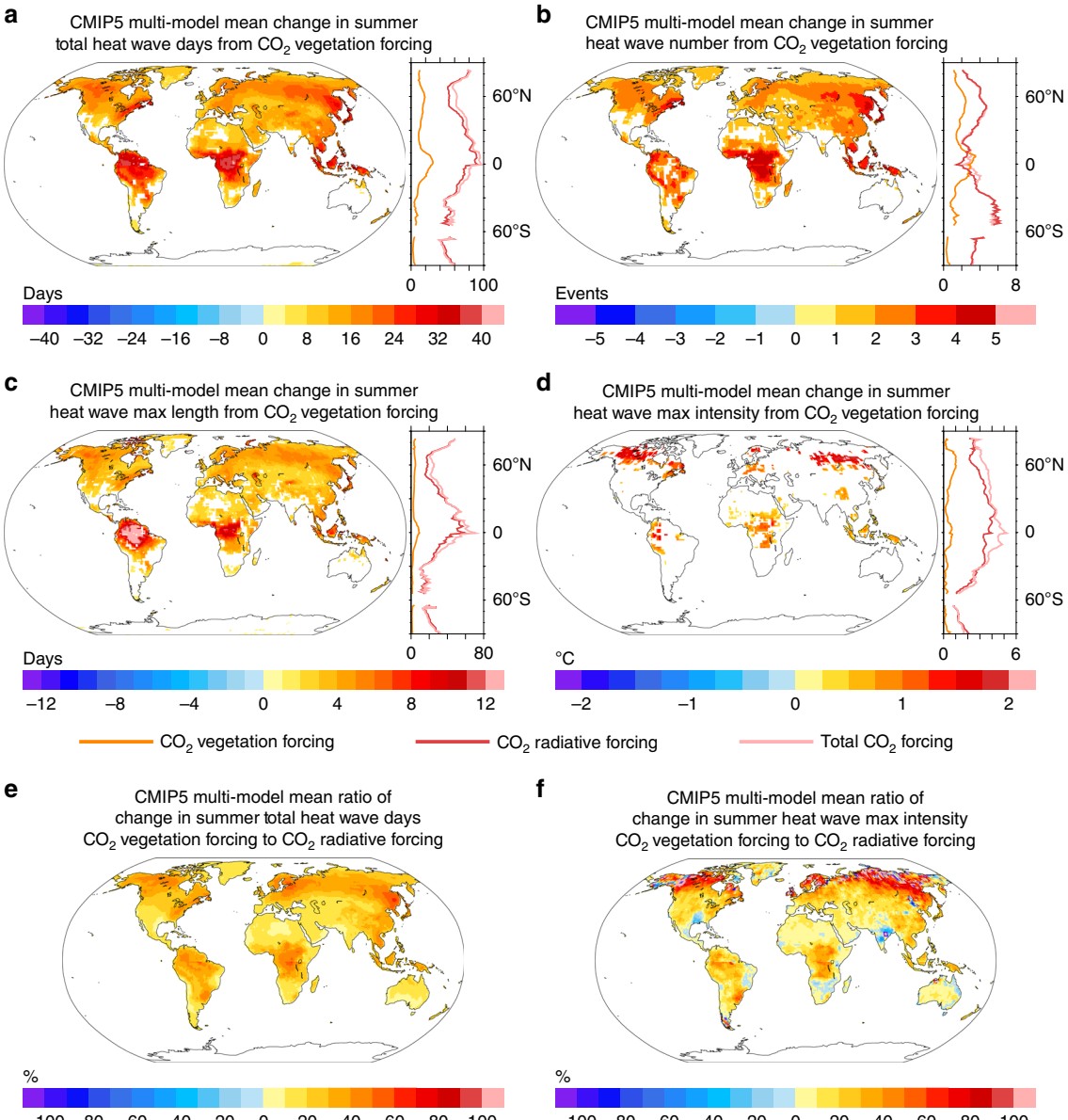

**Fig. 4** Change in heat wave metrics from $CO_2$ vegetation forcing. Multi-model mean change in summer **a** total heat wave days (HWTD), **b** heat wave number (HWN), **c** heat wave maximum length (HWML), and **d** heat wave maximum intensity (HWMI) from $CO_2$ vegetation forcing (maps). Multi-model mean zonal averages of the change in each heat wave metric from (orange) $CO_2$ vegetation forcing, (red) $CO_2$ radiative forcing, and (pink) total $CO_2$ forcing. Only land grid points are used in the zonal averages. Multi-model mean ratio (percentage) of the change in **e** HWTD and **f** HWMI from $CO_2$ vegetation and $CO_2$ radiative forcing ([$CO_2$ vegetation / $CO_2$ radiative] × 100). Only statistically significant changes are plotted in **a**-**d** (Methods). June–September (December–March) values are used in the Northern (Southern) Hemisphere. Plots represent 30-year averages

forcing and $CO_2$ fertilization are those located in warm, semi-arid, and arid climates (Fig. 2a, and Supplementary Figs. 2s-x and 3a-f). In these areas, models project substantial positive percent changes in LAI from $CO_2$ fertilization that counteract the impact of reduced stomatal conductance from $CO_2$ physiological forcing on transpiration (Supplementary Fig. 3a-f). These model projections are consistent with gas exchange theory and recent satellite observations that show the greatest impact of $CO_2$ fertilization in warm, arid climates where water is the dominant limiting factor for growth[23].

The CMIP5 results do not support the hypothesis that $CO_2$-vegetation forcing will reduce future mid-latitude summer heat waves[12]. This hypothesis posits that reductions in spring transpiration from $CO_2$ physiological forcing increase the summer soil moisture available to cool surface temperatures. We find that $CO_2$ vegetation forcing does increase spring season soil moisture in some regions in several of the CMIP5 ESMs (Fig. 5 and Supplementary Fig. 9). However, in the warm and arid regions where summer transpiration does increase slightly, the models do not project statistically significant reductions in summer heat waves (Fig. 4 and Supplementary Figs. 6–7). In most areas, the potential increase in transpiration from greater soil moisture and greater LAI does not balance the larger transpiration reduction induced by lower summer stomatal conductance, resulting in net decreases in summer transpiration and greater heat wave day frequency. In the presence of high $CO_2$, reduced stomatal conductance and transpiration appears to control soil moisture content in the CMIP5 models, rather than vice versa. The degree to which soil moisture influences model transpiration, often through a soil moisture stress parameter, is clearly important for

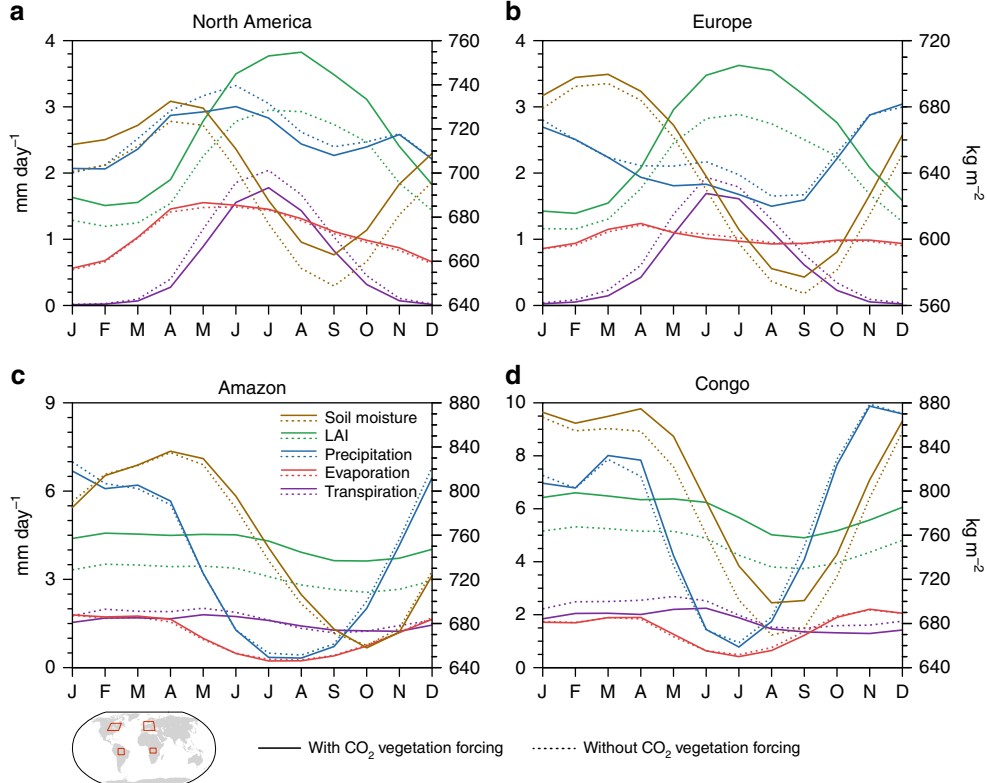

**Fig. 5** Annual cycle of vegetation and hydrologic variables in simulations with and without future $CO_2$ vegetation forcing. CMIP5 multi-model mean area-weighted averages of total-column soil moisture (brown), leaf area index (LAI) (green), precipitation (blue), evaporation (red), and transpiration (purple) for portions of **a** North America (40°N–55°N, 100°W–70°W), **b** Europe (40°N–60°N, 0°–30°E), **c** the Amazon basin (0°–15°S, 65°W–50°W), and **d** the Congo basin (0°–10°S, 15°E–30°E). Dashed (solid) lines represent the final 30 years of the *RadCO2* (*TotalCO2*) simulation. LAI (unitless), precipitation, evaporation, and transpiration correspond to the left *y*-axis. Soil moisture corresponds to the right *y*-axis. Output from HadGEM2-ES is not included in the plots

future projections of surface hydrology and extreme heat, and should be a focus of future land model development[38,41].

Reduced transpiration from $CO_2$ physiological forcing also initiates a number of climate system feedbacks that further enhance the likelihood and intensity of extreme heat events. In many mid and high latitude locations, a shift from latent to sensible heating (Fig. 3a) dries and stabilizes the boundary layer through lower ET and enhanced planetary boundary layer heights[42]. These regions experience fewer clouds, less rainfall, and greater surface solar insolation (Fig. 5a, b and Supplementary Fig. 4m-x). These changes lead to faster heat wave onset, enhanced heat wave temperatures, and longer heat wave events (Fig. 4). Future studies with access to daily-scale transpiration, winds, and geopotential heights should assess whether $CO_2$ vegetation forcing also alters the characteristics of atmospheric patterns that promote heat wave events.

Previous work that suggests $CO_2$ vegetation forcing will mitigate future mid-latitude heat waves utilizes a one-way nested regional climate model, which cannot capture these key climate system feedbacks[12]. Indeed, the regional climate model study uses the same land surface model (ORCHIDEE)[43] as the one employed in the IPSL-CM5A-LR model from CMIP5, which exhibits some of the largest transpiration reductions coupled with heat wave increases in response to $CO_2$ vegetation forcing in the mid and high latitudes due to such feedbacks. As demonstrated with the global ESMs in this study, when assessing the potential impact of $CO_2$ vegetation forcing on future heat wave events, it is necessary to consider locally and remotely driven climate system feedbacks, including cloud and precipitation changes, which may contribute to the climate response. It is also important to note

that the regional climate modeling work in ref.[12] analyzed a particularly long and severe heat wave event (the European summer 2003 heat wave). While it is clear that $CO_2$ vegetation forcing enhances the intensity and frequency of future heat wave events within CMIP5 models in general, it is possible that $CO_2$ vegetation forcing may result in greater surface latent cooling and reduced temperatures during one of these anomalously long heat waves. ESMs from CMIP5 tend to underestimate the frequency of the most severe heat waves[44], and therefore may not be suited to fully assess the impact of $CO_2$ vegetation forcing on all types of heat wave events.

Consideration of the vegetation response to rising $CO_2$ helps to explain the spatial patterns and intermodel differences of projected future heat extremes. Across the subset of CMIP5 models analyzed in this study, those that most severely underestimate the annual ratio of transpiration to ET in the present-day also project the smallest changes in transpiration in response to $CO_2$ vegetation forcing (Fig. 2b). The models with the largest transpiration reductions from $CO_2$ vegetation forcing project the greatest increases in summer warming and heat wave days (Supplementary Figs. 2s-x, 5, and 6a-f). These results point to the important role of existing vegetation parameterizations in shaping projections of extreme heat events. Data-model intercomparisons, such as the Free-Air $CO_2$ Enrichment (FACE) Model-Data Synthesis project[41] have identified a number of key processes responsible for model divergence in the simulation of transpiration, including the impact of soil moisture availability on stomatal conductance, the coupling between transpiration and canopy conductance, and the role of nutrient limitations. Though all models analyzed in this study project transpiration reductions and subsequent heat

wave increases due to $CO_2$ vegetation forcing, refinement of these model processes may help to constrain uncertainties in future heat wave projections. It is also important to note that several models from the larger CMIP5 ensemble (none of which are analyzed in this study) do not include the physiological effects of $CO_2$, as stomatal conductance in those models does not depend on $CO_2$[45]. Our results suggest that projections from those models would underestimate future changes in extreme heat. Ensuring that all models include the dependence of stomatal conductance on $CO_2$ concentration may help to constrain future changes in heat extremes.

More broadly, the substantial role of vegetation physiology in shaping future simulated hydrology and surface energy fluxes in ESMs highlights the need to develop mechanistic models of plant growth and physiology and to increase observational efforts toward understanding vegetation's role in the hydrologic cycle. Presently, models use semi-empirical formulations of stomatal conductance that do not capture the full range of stomatal behavior across plants (e.g., ref.[31]). Similarly, models struggle to simulate observed relationships between elevated $CO_2$ and changes in LAI[27]. Both of these factors limit confidence in projections of regional climate change, such as those presented in this study, and point to the need for increased process-based understanding and mechanistic models of stomatal conductance and carbon allocation in ESMs[46,47].

In terms of observations, the measurements of transpiration at large spatial scales are difficult to attain, and despite recent insights from stable isotope techniques[48] and satellite retrievals of LAI[39], estimates remain uncertain, making model-observation comparisons challenging. Additionally, a lack of FACE experiments in the tropics and high latitudes limits the ability to evaluate the modeled response of vegetation to elevated $CO_2$ in these regions. Future expansion of FACE experiments across biomes[49] will provide necessary validation of the large transpiration changes projected for the tropics and high latitudes (Fig. 1h and Supplementary Figure 2s-x). Model-data comparisons such as the FACE Model-Data Synthesis Project, which compare FACE data with output from land models forced with observed atmospheric conditions consistent with the FACE locations are a promising way forward for model evaluation[41].

The CMIP5 experiments analyzed in this study suggest transpiration plays an important role in diminishing extreme heat events in the present climate. In a high-$CO_2$ world, model reductions in transpiration from $CO_2$ physiological forcing outpace potential increases in transpiration from $CO_2$ fertilization leading to widespread heat wave increases. Although temperatures and heat extremes will continue to increase after anthropogenic $CO_2$ emissions cease due to thermal inertia in the oceans[50], the results here suggest the stabilization of atmospheric $CO_2$ concentrations will have the immediate benefit of limiting further reductions in transpiration from $CO_2$ physiological forcing, mitigating vegetation's role in enhancing extreme heat events. Given the potential for vegetation changes to shape future surface energy and water fluxes, improved process-based understanding and model representation of the role of vegetation in the carbon and hydrologic cycles is needed to prepare for and mitigate the acute impacts of future heat extremes.

## Methods

**Model data and experimental design**. We analyze simulations from six ESMs archived as part of the carbon-climate feedback experiment within CMIP5 (Table 1, Supplementary Table 1)[51]. The six ESMs are BCC-CSM1-1[52], CanESM2[53], CESM1-BGC[54], IPSL-CM5A-LR[55], HadGEM2-ES[56], and MPI-ESM-LR[57], and are chosen based on the availability of daily-scale temperature data needed for the analysis of heat extremes. For each model we assess the impact of $CO_2$ vegetation forcing on climate by comparing two simulations: one simulation that includes the fully-interactive radiative, physiological, and fertilization effects of

increasing $CO_2$ (*TotalCO2*, denoted *1pctCO2* in CMIP5, see Table 1) and one simulation that includes only the radiative effects of increasing $CO_2$ (*RadCO2*, denoted *esmFdbk1* in CMIP5, see Table 1). For our purposes, the only difference between the two sets of simulations is whether the vegetation in the model is directly influenced by the increasing $CO_2$—in *RadCO2* it is not. Additionally, we assess the impact of $CO_2$ radiative forcing on climate by comparing the set of *TotalCO2* simulations to a set of simulations that includes only the physiological and fertilization effects of elevated $CO_2$ (*VegCO2*, see Table 1), (denoted *esmFix-Clim1* in CMIP5). For our purposes, the only difference between the two sets of simulations is whether the atmospheric radiative transfer scheme is directly influenced by the increasing $CO_2$—in *VegCO2* it is not. In all three sets of simulations, $CO_2$ concentrations increase by 1% per year for 140 years, starting at 284 ppm and ending at about 1132 ppm. For reference, $CO_2$ concentrations in the high emissions RCP8.5 scenario are roughly 935 ppm in the year 2100[59]. Concentrations of aerosols and non-$CO_2$ greenhouse gases (other than water vapor) are fixed at preindustrial levels throughout each of the three 140-year simulations.

Our principal focus is on quantifying the impact of end-of-21$^{st}$-century $CO_2$ vegetation forcing on heat waves relative to end-of-21$^{st}$-century $CO_2$ radiative forcing. We calculate the impact of end-of-21$^{st}$-century $CO_2$ vegetation forcing on climate by subtracting the final 30 years of data in *RadCO2* (end-of-21$^{st}$-century radiative $CO_2$ forcing) from the final 30 years of data in *TotalCO2* (end-of-21$^{st}$-century total $CO_2$ forcing, see Table 2). Likewise, we calculate the impact of end-of-21$^{st}$-century $CO_2$ radiative forcing on climate by subtracting the final 30 years of data in *VegCO2* (end-of-21$^{st}$-century vegetation $CO_2$ forcing) from the final 30 years of data in *TotalCO2* (end-of-21$^{st}$-century total $CO_2$ forcing, see Table 2). We choose to analyze the final 30 years of each simulation (average $CO_2$ concentrations ~984 ppm) in order to assess modeled $CO_2$ vegetation forcing and $CO_2$ radiative forcing that are generally consistent with a high emissions scenario projection (RCP8.5) for the end of the 21$^{st}$ century[58].

We emphasize that we use the *RadCO2* simulations to back out the influence of $CO_2$ vegetation forcing on climate (i.e., *TotalCO2 – RadCO2*) rather than using the *VegCO2* simulations directly (and vice versa for $CO_2$ radiative forcing) because it allows for an assessment of $CO_2$ vegetation forcing ($CO_2$ radiative forcing) relative to future $CO_2$ radiative forcing ($CO_2$ vegetation forcing).

Our focus is on the summer season in each hemisphere (June–September (JJAS), in the Northern Hemisphere, December–March (DJFM) in the Southern Hemisphere), when mean temperatures and heat extremes reach their maximum intensity. We use the permutation test to assess the statistical significance of the differences between the simulations at the 95% confidence level within each individual model, and present multi-model mean figures to show model agreement. Within the multi-model mean figures, a grid box is filled with the mean value of all six ESMs when at least four of the six models exhibit statistically significant changes in the same direction as the multi-model mean change at that grid point. When a location does not meet these criteria, the grid box is left unfilled (white). Maps for each individual model are provided as Supplementary Figures. The HadGEM2-ES model does not archive evaporation or transpiration variables and is therefore not in the multi-model mean plots for those variables; we explicitly noted this when we use HadGEM2-ES ET as a fill-in for evaporation or transpiration.

**Heat wave detection**. Heat waves are detected following[59] and the indices for monitoring temperature extremes put forth by the World Meteorological Organization (WMO) Commission for Climatology (CCl)/Climate Variability and Predictability (CLIVAR)/Joint Technical Commission for Oceanography and Marine Meteorology (JCOMM) Expert Team on Climate Change Detection and Indices (ETCCDI)[60]. Specifically, a heat wave is defined as an event of at least three consecutive days during which the daily maximum temperature exceeds the calendar day 90$^{th}$ percentile value from a reference period, based on a 5-day moving average. A percentile value for each calendar day is used to account for seasonality, and a 5-day moving average is used to account for temporal autocorrelation in the daily data[59].

We used four heat wave metrics to characterize changes in extreme heat events: The HWTD is the sum of all days that meet the heat wave criteria each season. The heat wave maximum length (HWML) is the length in days of the longest heat wave event each season. The heat wave number (HWN) is the average number of heat waves per season. The heat wave maximum intensity (HWMI) is the maximum daily temperature reached during each heat wave event during the season. All metrics are calculated for each year and then averaged over the 30-year time period.

The reference period used to calculate temperature percentiles and heat waves depends on the $CO_2$ forcing of interest. Given our experimental design (see above), the reference period for calculating changes in heat waves due to end-of-21$^{st}$-century $CO_2$ vegetation forcing is the final 30 years of *RadCO2*. The reference period for calculating changes in heat waves due to end-of-21$^{st}$-century $CO_2$ radiative forcing is the final 30 years of *VegCO2*.

**Sensitivity analyses**. We provide two sensitivity analyses for our results. First, to assess the influence of our chosen reference climate on changes in heat extremes, we also analyze the change in heat wave indices within (rather than across) model simulations by subtracting the first 30 years of data in the *VegCO2* (*RadCO2*)

simulation from the final 30 years of data in the *VegCO2* (*RadCO2*) simulation (Table 2). We also quantify the total CO2-driven (CO2 vegetation forcing + CO2 radiative forcing) response of heat extremes by subtracting the first 30 years of the *TotalCO2* simulation from the final 30 years of the *TotalCO2* simulation. Accordingly, in these analyses, the reference period temperature thresholds and heat waves are defined using the first 30 years of data in *VegCO2*, *RadCO2*, and *TotalCO2*.

Second, to assess the influence of different levels of CO2-forcing on our results, we also analyze the vegetation-driven responses of extreme heat metrics for CO2 concentrations consistent with the middle 21st century in a high emissions scenario (~575 ppm). To assess the impact of mid-century CO2 vegetation forcing we subtract the 30-year time period between years 58 and 87 of the *RadCO2* simulation from the corresponding 30-year time period in the *TotalCO2* simulation (Table 2). Simulation years 58–87 are chosen to reflect CO2 values that are roughly consistent with years 2040–2070 in the RCP8.5 pathway[58].

**Model vegetation description and validation**. CO2 physiological forcing is directly tied to the representation and response of stomatal conductance in the models. All six ESMs relate stomatal conductance to photosynthesis via semi-empirical formulations. Five of the six ESMs use either the Ball–Berry conductance model[61] or the Leuning conductance model[62] (Supplementary Table 1). The primary difference between the two formulations is their treatment of atmospheric moisture content: Ball–Berry uses relative humidity while Leuning uses vapor pressure deficit. In the sixth model, MPI-ESM-LR, the formulation of stomatal conductance does not include a dependency on atmospheric humidity[63,64]. All else equal, all models exhibit reductions in stomatal conductance in response to elevated atmospheric CO2 concentrations at the leaf surface based on their functional forms.

Calculations of LAI are prognostic in all ESMs and depend on plant carbon stocks (in part determined by CO2 fertilization), carbon allocation (the distribution of carbon to leaves, roots, and stems), and leaf turnover rates, as well as climate factors including temperature, soil moisture, and sunlight[37]. HadGEM2-ES and MPI-ESM-LR both include dynamic vegetation modules (Supplementary Table 1), which allow the communal assemblages of plant functional types to change, though inclusion of dynamic biogeography appears to have little effect on intermodel differences in projected LAIs[37]. We compare ESM simulations of LAI with the satellite-based AVH15C1 LAI dataset (years spanning 1980–2010) derived from the National Oceanic and Atmospheric Administration (NOAA) Climate Data Record (CDR) of Advanced Very High Resolution Radiometer (AVHRR) Surface Reflectance[65].

All data (observations and CMIP5) are interpolated to a common 1° × 1° grid using a patch recovery method[66].

**Data availability**. All CMIP5 data analyzed in the current study are publicly available on the Earth System Grid Federation website: https://esgf.llnl.gov/. Observed LAI data from the NOAA CDR of AVHRR Surface Reflectance are publicly available from the NOAA National Centers for Environmental Information (NCEI) website:
https://data.nodc.noaa.gov/cgi-bin/iso?id=gov.noaa.ncdc:C00898

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

## Acknowledgements

This work was supported by the Turner Postdoctoral Fellowship awarded to Christopher Skinner and by National Science Foundation Award 1602956. Justin S. Mankin was supported by The Earth Institute of Columbia University. We wish to thank Jürgen Knauer for helpful discussions. We acknowledge the World Climate Research Program and the climate modeling groups for making available their model output, and the U.S. Department of Energy's Program for Climate Model Diagnosis and Intercomparison for coordinating and supporting database development. This research was supported in part through computational resources and services provided by Advanced Research Computing at the University of Michigan, Ann Arbor. Lamont contribution number 8196.

## Author contributions

C.B.S. conceived of the study and conducted the analysis. C.B.S., C.J.P., and J.S.M. interpreted the results and wrote the paper.

## Additional information

**Competing interests:** The authors declare no competing interests.

