## [Peer Review File · Nature Communications]

Reviewers' comments:

Reviewer #1 (Remarks to the Author):

Skinner et al. examine the role of plant physiology in the amplification of heat extremes in responses to elevated CO₂. For future reference, line numbers would have been really helpful!

Overall, I think this study is thorough, well presented and largely, clearly written. I personally, didn't find anything particularly new here, but that isn't to say a wider audience won't read with interest.

I am however, a bit bothered in places with the claimed motivation, for example, "Previous work suggests that the combined effects of CO₂ on vegetation will enhance summer season transpiration and mitigate the occurrence of future heat waves". I wouldn't personally argue that was a robust claim. This would be very much dependent on vegetation type and where you were located. Can I refer the authors to a detailed treatment of this issue of model water savings (Medlyn et al. 2016). I think toning down these apparent motivations, or simply rephrasing might make the paper stronger.

The above matters, because they go on to argue in their abstract: "Contrary to expectation, the vegetation response to a quadrupling of CO₂ increases summer heat wave occurrence by 20 days or more—30-50% of the radiative response alone". I'm not clear whose expectation this is being based on? To me this seems entirely logical at the most fundamental level. If eCO₂ leads to reductions in *g_s*, in turn, this will manifest itself in reductions in transpiration. By how much, is very debatable in model world of course (see De Kauwe et al. 2013). But logically, if transpiration were reduced one would expect an increase in heat extremes. Again this all needs to be tested to see how robust such back of the envelope statements are. And this is what the authors of course do in this paper. However, by setting out these straw man arguments, it makes the results appear more unexpected than needs be? I think if they want to explore this counter position (that CO₂ = fewer heat extremes), then they need to be far more explicit about the assumptions behind those studies.

In terms of potential improvements. I'd ask the authors to consider making a plot of WUE and CO₂. It would be interesting to know how differently the model predict the WUE response to CO₂ to be? Given they focus on changes in transpiration, if the WUE was very different, then one might question how robust the modelled predictions of heatwaves indices were? Or alternatively, the authors could use this as a constraint to reduce the apparent spread in heatwave indices observed. I'd also ask a similar question about predicted precipitation. How variable is this across models? How do the authors propose (they haven't currently) separating the CO₂ effects on vegetation on heatwaves, from models differences in PPT on heatwaves?

Introduction:

- The authors highlight that CO₂ can lead to increased LAI. This is a very broad statement and is certainly not the prevailing hypothesis. I realise they state "can lead", but it is likely to only be certain ecosystems, or those at certain stages of canopy development (see Norby & Zak, 2011). This statement could be more precisely crafted.

- The authors cite reference 12 to support the assertion that CO₂-induced water savings could save sufficient water to cool the boundary layer later in the peak of summer. Are there any other studies that would also support this? At a fundamental level I agree, but I wonder how much additional water the authors think CO₂ would save the plants? And in turn, how long this additional source of water would humidify the boundary layer for? I'm not nitpicking here, I think this is a fundamental point and a stronger evidence line here would enhance the paper. They do cite a tentative link to fewer hot extremes in irrigated croplands. This is nice, but seems at least a

level removed. I guess I'm after something more quantitative here.

Methods:

- I'm not clear why the comparison point is the final 30 years i.e. the point at which CO₂ has hit ~1132 ppm - 30 years. There is no justification provided for why this choice is made. Given, this is a point so far into the future, I'm not immediately seeing why this point was chose (apart from it being the easiest to process)? Can the authors provide a physical justification?
- Calculation of LAI of course also depends on assumptions about turnover as well as the GPP fixed and the carbon allocation. So there are further reasons for models differences not given in the text. Turnover should be added to this sentence.

Results:

- Given the intended publication is nature communications, it doesn't make sense to me to start with a discussion about LAI as the starting point of the results? Surely that authors would start with their key result and work backwards from there? Just a thought. In fact, is the whole chunk of text about model skill at simulating LAI strictly necessary given the focus of the paper? At best, this is supplementary material? It doesn't really say anything relevant for a paper on heatwaves.
- Fig 1a/b: I think simply the average LAI change is meaningless? Would a zonal plot be more instructive?
- Fig 1c/d: I find it interesting that none of this suite of models seemingly predict that LAI changes cancel any stomatal reduction in g_s (expressed as transpiration) anywhere on the globe? Is that just true for the regions in which models agree? Or is this true for each of the models simulations? I find this a very surprising result and would certainly think it warrants some further investigation.
- I have to confess that I got very little out of Fig 2. I either didn't follow it, or it needs a bit of work to make it clearer. Fig 2b in particular?
- Please quantify this: "the reductions in transpiration from CO₂ vegetation forcing in the models are large enough". How big are the changes (? mm/summer)?

Discussion:

- I have no idea what this means: "Several models from CMIP5 do not include the plant physiological effects of CO₂", please clarify.

References:

- Norby & Zak (2011) Ecological Lessons from Free-Air CO₂ Enrichment (FACE) Experiments. *Annual Review of Ecology, Evolution, and Systematics*, 42, 181-203.
- Medlyn et al. (2016) Using models to guide field experiments: a priori predictions for the CO₂ response of a nutrient- and water-limited native Eucalypt woodland. *Global Change Biology*, 22, 2834–2851,

Reviewer #2 (Remarks to the Author):

This paper quantifies how the response of plants to rising CO₂ (both through leaf area and physiological changes) influences heat wave occurrence, length, and intensity. They find that heat waves are increased by plant responses, particularly in places with very high leaf area (lai) to

begin with, i.e. the tropics and mid and high latitudes with high lai. Even with increases in lai under high CO₂, the decreases in water flux due to stomatal closure are large enough to lead to higher maximum temperatures. These results disagree with prior work showing that in Europe, a 2003-type heat wave is alleviated by plant responses when they save water from earlier in the season due to reduced stomatal conductance allowing for higher transpiration during the peak of the heat wave and thus reductions in peak temperatures.

This analysis is a novel and useful contribution to our understanding of how the coupled Earth system responds to change. It helps to illustrate that plants play a role in many climate impacts, even if those influences are not always identified in coupled simulations and often de-facto attributed to the radiative effects of CO₂.

I think the paper is appropriate for publication in Nature Communications. I have two major comments. I also have a number of very minor comments.

Major comments:

1. This paper doesn't cite Sellers et al. 1996!

Sellers et al. 1996 establishes the framework and concept that this work is based on by showing that temperature is influenced by the physiological response of plants to rising CO₂. The authors cite many papers that are derived following Sellers, but not the original. The authors can obviously fix this, but it is an egregious error! It seems like this may be needed elsewhere in the introduction, such as the first paragraph, but at a minimum in the following two locations:

i. "due to the non-radiative effects of higher CO₂ is known as CO₂ physiological forcing (17)"

Sellers et al 1996 is the appropriate reference here. Betts follows on Sellers.

ii. "Meanwhile, CO₂ physiological forcing limits transpiration and enhances the ratio of sensible to latent heat fluxes at the leaf surface, increasing boundary layer temperatures (22)."

Also here, covered by Sellers far earlier than Cao.

2. There is no formal comparison of radiative effects on heat waves vs. vegetative effects on heat waves relative to the combined effect.

The relative sizes of radiative vs. vegetative effects was mentioned in the abstract (veg is 30-50% of radiative effect), but then not formally quantified elsewhere in the text. The parallel plots for radiative effects are shown in figure S6 and S7, but I would like to see something that quantifies the relative size of the two terms, and how they contribute to the total change in heat waves. This seems like a critical piece of information in assessing if the vegetative influence matters or not. The authors should have all information necessary to make this assessment. I feel that it is needed for the manuscript to tell a complete story. I realize that the radiative effects are larger, but it is important to know the relative contribution of the vegetation effects to the total response.

Minor Comments:

It would be very helpful if the authors provided line numbers, or at a bare minimum page numbers. As such, I have tried to give approximate locations for my comments and include some text for context. I assigned the page numbers starting with the first page of the document as page 1.

- The names of the six ESMs and references for each model need to be in the paper somewhere, either inline in the text or in a table. I expected to find them in Table 1 but was surprised that was only describing the experiments. I see the names of the models are in the supplemental table, but still not the references. I think the references should be in the main reference list.

- (beginning of results section) These simulations do not include land use, so we shouldn't expect them to exactly capture observed LAI even in the present day. Perhaps this should be mentioned.

- (end of page 10, beginning of page 11) "positive" cloud cover is confusing. Could the authors use "increasing" instead?

- (top of page 12) "In comparison to the pattern of heat wave day changes from CO₂ radiative forcing, CO₂ vegetation forcing primarily impacts wetter, vegetated areas of the mid and high latitudes (Figs. 4A and S6A-L)." It would be helpful to move the figure reference (S6A-L) to immediately follow this first statement since that is where that quantity is shown.

- (middle of page 14) "slightly greater total-column soil moisture at the start of the summer season in most models (see Table S1 for the hydrologically active soil column depths in each model)" How did the authors define "hydrologically active" soil column depth that differs from total soil column depth? CESM1-BGC, for instance, has a depth of 5m, but is listed in the table as 2.86m for hydrologically active. I can't find an explanation of how this was determined and I don't find it obvious.

References:

P. Sellers, L. Bounoua, G. Collatz, D. Randall, D. Dazlich, S. Los, J. Berry, I. Fung, C. Tucker, C. Field, and T. Jensen. Comparison of radiative and physiological effects of doubled atmospheric CO₂ on climate. *Science*, 271(5254):1402–1406, Mar. 1996.

Reviewer #3 (Remarks to the Author):

Overall Review

The manuscript addresses an important topic: how changes in vegetation LAI and stomatal physiology induced by elevated CO₂ alter land-atmosphere feedbacks and precisely heat waves. The authors analyze results from CMIP5 model ensemble and specifically simulations that include the role of CO₂ only on the radiative component of the models (radiative forcing) and only on vegetation component (physiological and fertilization forcing) (Table 1). In this way, they can separate the two effects. The main findings are that decreased stomatal conductance during summer generally reduces transpiration despite an increase in LAI (Fig. 2), which leads to an enhancement of future heat-waves frequency and intensity (Fig 4). As far as I know, only a couple of studies have looked at similar questions (Leomordant et al 2016; Kala et al 2016) but they had a more regional focus and do not analyze Earth System Model projections in such systematic way. Furthermore, the results of the current article are also different from results in Leomordant et al 2016. Overall, despite some concern on the plant-physiology implemented in CMIP5 Earth System Models, I think the study is well carried-out, well presented, and results are interesting and important. In summary, I enjoyed reading this research. However, in my opinion, there are a couple of aspects that need more discussion,

(i) One of the major concern I have is related to the stomatal conductance parameterization used in the various ESMs (Table S1). This is not something the authors can change but deserves a bit more attention in the discussion. Current ESMs use static parameters and employ a stomatal model that forcefully close stomata considerably in response to CO₂. This is a main driver of all the results presented in the article. Now, there is evidence that this is happening for a large number of species, but there is also evidence that it does not happen for some other species (e.g., Field et al 1995; Medlyn et al 1999; Keel et al 2007). In other words, the adopted stomatal parameterization

is rather empirical and not flexible enough to accommodate for a range of real observed behaviors. This has been already discussed before in other articles (Damour et al, 2010 Paschalis et al 2017) but needs an explicit mention because it can affect most of the results. It is not just a parameterization problem (e.g., Kala et al 2016) as the authors also discusses in this paper but likely a structural model problem. Furthermore, the response of LAI to increase CO2 is also quite difficult to simulate with current ESMs (see discussion in Fatichi et al 2016), which adds an additional level of uncertainty that needs to be explicitly mentioned.

(ii) The study of Leomordant et al 2016 is based on actually observed summer of 2003 in Europe and some of the differences may be also related to the persistence of the heatwave in that summer. In other words if the period interested by heat-waves is short, temperature may be amplified by stomatal closure, however if the period is long the saved water can be used later in the summer to decrease sensible heat and temperature in other heat-waves that can occur later on. If ESMs do not capture well the current length, correlation and intensity of heatwaves, these effects may not be so evident in the analyzed simulations. ESMs may have issues in reproducing the most persistent weather patterns and therefore the most-extreme heat-waves also for present climate conditions. I wonder if a comparison of observed and simulated heat-wave statistics has been published previously and can be referred to, or if such a comparison can be made here and will strengthen the relevance of this manuscript. Another way, could be to look if the physiological effect is becoming less important as the dry season progresses, especially in water-limited climates. As a matter of fact, the most clear physiological effect on heatwave statistics is on the humid tropics (Fig. 4), which are less likely water-limited.

(iii) Most of the results are presented and discussed for the last 30 years of the analyzed simulations that have a CO2 concentration of roughly 1000 ppm. It would be interesting to have in the article some number as a reference (maybe some Figure in the Supp. Information) also for CO2 concentrations of 500-600 ppm that are what we expect in the near future conversely to a far future.

Sincerely,

Simone Fatichi

SPECIFIC COMMENTS

Page 3. Line 3. I am not sure the meaning of "human morbidity" is generally known.

Page 3. Line 19. I would suggest to use "reduced" rather than "narrowed".

Page 4. Line 8-10. and Page 5 Line 6-10. These statements are strictly true only if there is enough water in the soil that plants are never water stressed. If this is not the case, CO2 can temporary reduce transpiration but the saved water will be used anyhow during the growing-season and the integrated transpiration and evapotranspiration will be similar in any CO2 or LAI scenarios. In a severely water limited region evapotranspiration is not a function of CO2 or LAI but of the amount of precipitation regardless of CO2 levels (e.g., Fatichi et al 2016).

Page 15. "maintaining the same transpiration" rather than "limiting transpiration reduction".

Page 18. Line 1. Leomordant et al. 2016 also look to a very peculiar heat-wave in western Europe -the summer of 2003 - with vegetation that experience water limitations, at least in their simulations. This can be an additional reason to explain the difference. In any case, for the first part of the summer results agree with the CMIP5-ESMs results.

Caption of Figure 2. "Median grid point % change in LAI for locations with" is a bit awkward. If I understood correctly I would write "Median change in LAI [%] over the grid points with ..."

Caption of Figure 3. The evaporative fraction is defined as the ratio of latent heat flux to the sum of latent and sensible heat fluxes and not as sensible heat over latent heat, which is the Bowen Ratio. Please correct. I guess, in the figure, you show changes in either the real evaporative fraction or the inverse of the Bowen ratio.

Figure S8. The small map of the world contains polygons only for two selected regions rather than four.

References

Damour, G, Simonneau T, Cochard H, Urban L (2010). An overview of models of stomatal conductance at the leaf level. *Plant Cell Environ* 33:1419–1438.

Fatichi S, S Leuzinger, A Paschalis, JA Langley, A Donnellan-Barracough, and M Hovenden (2016b). Partitioning direct and indirect effects reveals the response of water limited ecosystems to elevated CO₂. *PNAS, USA*. 113(45) 12757-12762, doi: 10.1073/pnas.1605036113

Field CB, Jackson RB, Mooney HA (1995) Stomatal responses to increased CO₂: implications from the plant to the global scale. *Plant, Cell and Environment* 18(10):1214–1225

Kala J, et al. (2016) Impact of the representation of stomatal conductance on model projections of heatwave intensity. *Sci Rep* 6:23418.

Keel, SG., Pepin, S, Leuzinger S, Körner C (2007). Stomatal conductance in mature deciduous forest trees exposed to elevated CO₂. *Trees* 21, 151–159.

Lemordant L, et al. (2016) Modification of land atmosphere interactions by CO₂ effects: Implications for summer dryness and heat wave amplitude. *Geophysical Research Letters* 43(19):10,240-210,248

Medlyn, B.E., et al., 1999. Effects of elevated [CO₂] on photosynthesis in European forest species: a meta-analysis of model parameters. *Plant Cell Environ*. 22(12), 1475–1495

Paschalis A, et al. (2017). On the variability of the ecosystem response to elevated atmospheric CO₂ across spatial and temporal scales at the Duke Forest FACE experiment. *Agricultural and Forest Meteorology*, 232, 367-383.

Response to Reviewers Document for “Amplification of Heat Extremes by Plant CO₂ Physiological Forcing”, by Skinner, Poulsen, and Mankin.

We thank the reviewers for their insightful comments, which have greatly improved the manuscript. Below we detail the changes that we made to the manuscript in response to each of the individual reviewer comments. While there is some duplication of material throughout the document, our intention is that our comprehensive comment-by-comment explanation helps to easily and efficiently evaluate exactly how each individual comment has been addressed.

The reviewer comments are shown in **bold text**. Our responses are shown in plain text. Quotations from our revised text are shown in *italics*.

Reviewer 1 Comments

Skinner et al. examine the role of plant physiology in the amplification of heat extremes in responses to elevated CO₂. For future reference, line numbers would have been really helpful!

We apologize for this frustrating oversight and we appreciate your efforts to provide clear and organized feedback in the absence of line numbers. We have added page numbers and line numbers to the revised manuscript.

Overall, I think this study is thorough, well presented and largely, clearly written. I personally, didn't find anything particularly new here, but that isn't to say a wider audience won't read with interest.

Thank you for this positive evaluation.

I am however, a bit bothered in places with the claimed motivation, for example, "Previous work suggests that the combined effects of CO₂ on vegetation will enhance summer season transpiration and mitigate the occurrence of future heat waves". I wouldn't personally argue that was a robust claim. This would be very much dependent on vegetation type and where you were located. Can I refer the authors to a detailed treatment of this issue of model water savings (Medlyn et al. 2016). I think toning down these apparent motivations, or simply rephrasing might make the paper stronger.

The above matters, because they go on to argue in their abstract: "Contrary to expectation, the vegetation response to a quadrupling of CO₂ increases summer heat wave occurrence by 20 days or more—30-50% of the radiative response alone". I'm not clear whose expectation this is

being based on? To me this seems entirely logical at the most fundamental level. If eCO₂ leads to reductions in gs, in turn, this will manifest itself in reductions in transpiration. By how much, is very debatable in model world of course (see De Kauwe et al. 2013). But logically, if transpiration were reduced one would expect an increase in heat extremes. Again this all needs to be tested to see how robust such back of the envelope statements are. And this is what the authors of course do in this paper. However, by setting out these straw man arguments, it makes the results appear more unexpected than needs be? I think if they want to explore this counter position (that CO₂ = fewer heat extremes), then they need to be far more explicit about the assumptions behind those studies.

We agree that the language we used in the abstract to motivate our study relied too heavily on the results of one paper (Lemordant et al., 2016, GRL) and as such did not reflect a large body of work that suggests the vegetation response to elevated CO₂ may in fact enhance future heat extremes (though this hasn't been shown before).

As suggested, we have rephrased the abstract language to more fully represent the existing body of work on this topic. We have also removed the phrase "Contrary to expectation".

The abstract now says:

Plants influence extreme heat events by regulating land-atmosphere water and energy exchanges. The contribution of plants to changes in future heat extremes will depend on the responses of vegetation growth and physiology to the direct effects of elevated CO₂. Here we use a suite of earth system models to disentangle the radiative versus vegetation effects of elevated CO₂ on heat wave characteristics. Vegetation responses to a quadrupling of CO₂ increase summer heat wave occurrence by 20 days or more—30-50% of the radiative response alone—across tropical and mid-to-high latitude forests. These increases are caused by CO₂ physiological forcing, which diminishes transpiration and its associated cooling effect, and reduces clouds and precipitation. In contrast to recent suggestions, our results indicate CO₂-driven vegetation changes enhance future heat wave frequency and intensity in most vegetated regions despite transpiration-driven soil moisture savings and increases in aboveground biomass from CO₂ fertilization.

In terms of potential improvements. I'd ask the authors to consider making a plot of WUE and CO₂. It would be interesting to know how differently the model predict the WUE response to CO₂ to be? Given they focus on changes in transpiration, if the WUE was very different, then one might question how robust the modelled predictions of heatwaves indices were? Or alternatively, the authors could use this as a constraint to reduce the apparent spread in heatwave indices observed. I'd also ask a similar question about predicted precipitation. How variable is this across models? How do the authors propose (they haven't currently) separating the CO₂ effects on vegetation on heatwaves, from models differences in PPT on heatwaves?

Thank you for these helpful suggestions. We now include a plot of the percent change in WUE as Supplementary Figure 3g-l. We find that the models do in fact show agreement on the change in WUE. WUE increases throughout all biomes and the percent change in WUE is particularly large in semi-arid regions that exhibit the largest percent increases in LAI. For the most part, the largest increases in WUE also correspond with the regions that exhibit increases in summer mean transpiration. This highlights the fact that increases in WUE do not necessarily imply a reduction in ecosystem water use. We now include a brief discussion of the percent WUE changes in the Results Section.

Specifically, we have added new text that states:

Summer season water use efficiency (WUE), defined as the ratio of summer gross primary productivity (GPP) to summer transpiration, increases across all biomes (Supplementary Fig. 3g-l). The percent change in WUE is particularly large in the relatively warm semi-arid regions that exhibit mean increases in transpiration in response to CO₂ vegetation forcing (Supplementary Fig. 3g-l). This result highlights the important point that increases in WUE do not necessarily imply a reduction in total plant water use if photosynthesis and biomass increase (38). - Page 6, Lines 188-194

As suggested, we now include a plot of the change in summer mean precipitation as Supplementary Figure 4s-x. For space considerations, we have replaced the figure showing the change in summer downward surface shortwave radiation (the previous Supplementary Figure 4s-x) with the change in summer precipitation. The change in summer downward shortwave radiation did not add much additional information beyond that which the plot of mean summer cloud cover change provides (Supplementary Figure 4m-r).

The changes in mean precipitation are, for the most part, consistent with the changes in cloud cover (Supplementary Figure 4m-r). Reductions in precipitation surely contribute to increases in temperature and heat wave days in the Northern Hemisphere mid to upper latitudes. In the tropics, several models project increases in precipitation in response to CO₂ vegetation forcing (as discussed in Skinner et al. 2017). These increases in tropical precipitation limit the amount of surface warming in regions such as Africa and parts of western South America.

Your point about separating out the impact of the direct CO₂ vegetation effects on heat waves from the indirect CO₂ vegetation-driven precipitation effects on heat waves is a good one. However, in the context of this manuscript, we consider the precipitation response to be a component of the full CO₂ vegetation forcing. Indeed, we suggest that it is critical to consider the full suite of changes in climate system processes (such as changes in rainfall) from CO₂ vegetation forcing, in order to fully understand how the vegetation response to CO₂ will impact heat extremes.

Introduction:

- The authors highlight that CO₂ can lead to increased LAI. This is a very broad statement and is certainly not the prevailing hypothesis. I realise they state "can lead", but it is likely to only be certain ecosystems, or those at certain stages of canopy development (see Norby & Zak, 2011). This statement could be more precisely crafted.

We have changed the text to reflect the fact that the response of LAI to CO₂ will depend on factors including the stage of plant development and nutrient, light and water limitations.

Specifically, the text now states:

While the magnitudes of the fertilization and physiological responses vary by plant species and under different environmental conditions, including water, light, and nutrient availability, observational and modeling evidence suggests that the two effects have opposing influences on climate (19, 20). Though enhanced CO₂ often has a limited impact on leaf area index (LAI, defined as one-sided leaf area per unit ground surface area), in nutrient-limited regions, and in mature forests (21), CO₂ fertilization can lead to enhanced LAI during the early stages of plant development (21, 22) and in regions that are water-limited (23). Greater LAI can enhance plant transpiration and surface evaporative cooling given sufficient moisture supply (24, 25). - Page 4, Lines 89-97

- The authors cite reference 12 to support the assertion that CO₂-induced water savings could save sufficient water to cool the boundary layer later in the peak of summer. Are there any other studies that would also support this? At a fundamental level I agree, but I wonder how much additional water the authors think CO₂ would save the plants? And in turn, how long this additional source of water would humidify the boundary layer for? I'm not nitpicking here, I think this is a fundamental point and a stronger evidence line here would enhance the paper. They do cite a tentative link to fewer hot extremes in irrigated croplands. This is nice, but seems at least a level removed. I guess I'm after something more quantitative here.

The modeling study by Lemordant et al. (2016) (reference 12) was motivated by observations from the Swiss Canopy Crane Project (a FACE site in a temperate forest in Switzerland). Keel et al. (2007) compared stomatal conductance in species exposed to ambient and elevated CO₂ treatments during the 2003 European summer heat wave at the Swiss FACE site. They found that some plant species exposed to elevated CO₂ exhibited greater transpiration near the end of summer compared with their species counterparts in ambient CO₂ conditions. They related this increase in transpiration at the end of the heat wave to the observed CO₂-induced soil moisture savings from the earlier portion of the summer. We now cite the paper by Keel et al. (2007).

Additionally, Field et al. (1997) showed that reduced growing-season transpiration from elevated CO₂ in a grassland ecosystem led to greater transpiration during the latter part of the observation period compared with the same grassland ecosystem growing under ambient CO₂. We now cite Field et al. (1997).

We agree that including examples of observed water savings from elevated CO₂ experiments will strengthen the line of reasoning. We now provide observed estimates of water savings from two elevated CO₂ studies: one in a grassland ecosystem (Field et al. 1997) and one in a temperate deciduous tree species (Warren et al. 2011). The objective here was to choose two studies that did in fact show statistically significant reductions in transpiration from elevated CO₂ treatments.

We have modified the text to state:

Given the impact of CO₂ physiological forcing and CO₂ fertilization (hereafter collectively referred to as CO₂ vegetation forcing) on surface moisture and energy fluxes, each may contribute to projected changes in future heat wave events, but the net effects on projected heat waves remain unresolved. The vegetation response to elevated CO₂ could mitigate the frequency and intensity of summer heat waves by increasing the springtime canopy water use efficiency via CO₂ physiological forcing, thus increasing the soil water available for evapotranspirative cooling later in the peak of summer (12, 30, 31). For example, observed growing season water savings from CO₂-induced transpiration reductions range from up to ~2.15 mm day⁻¹ in some grassland ecosystems (exposed to 100% increase in CO₂) (30), to ~0.6 mm day⁻¹ in some temperate deciduous tree species (exposed to 40% increase in CO₂) (22). Coupled with increasing summertime LAI via CO₂ fertilization, regions with greater soil moisture, particularly those that would have otherwise become water-limited during the summer, could see increases in transpiration and therefore surface cooling when transpiration demand is high (such as during dry, hot days) (12). Indeed, observational and modeling work shows that irrigated cropland has experienced fewer summer hot extremes in recent decades as a result of enhanced ET from greater soil moisture (32, 33). - Pages 4-5, Lines 108-124

Methods:

- I'm not clear why the comparison point is the final 30 years i.e. the point at which CO₂ has hit ~1132 ppm - 30 years. There is no justification provided for why this choice is made. Given, this is a point so far into the future, I'm not immediately seeing why this point was chose (apart from it being the easiest to process)? Can the authors provide a physical justification?

Thank you for pointing this out. We chose to analyze the final 30 years of the simulations for a couple of reasons. First, as you noted, this is a point in the simulations where a clear signal has emerged. Second, the CO₂ during the final 30 years of the simulation, though high, is generally consistent with an end of 21st century high emissions scenario (year 2100 values of CO₂ in the RCP8.5 simulation are about 935 ppm). Our analysis therefore provides insights into the role of CO₂ vegetation forcing in shaping the projected changes in heat extremes consistent with the end-of-century forcing from a

business-as-usual greenhouse gas emissions scenario. Third, we wanted to compare the CMIP5 results with the results from Lemordant et al. (2016), who analyzed the impact of CO₂ vegetation forcing under high CO₂ concentrations of 936 ppm.

As suggested by Reviewer 3, we now include additional analysis of CO₂ vegetation forcing from a 30-year time period with CO₂ concentrations between 500 and 668 ppm (simulation years 58 – 87). This range of CO₂ roughly corresponds to years 2040 – 2070 in the RCP8.5 scenario. These results can be seen in Fig. S8.

We have added new text that states:

In all three sets of simulations, CO₂ concentrations increase by 1% per year for 140 years, starting at 284 ppm and ending at about 1132 ppm. For reference, CO₂ concentrations in the high emissions RCP8.5 scenario are roughly 935 ppm in the year 2100 (58). - Page 21, Lines 483-486

Our principal focus is on quantifying the impact of end-of-21st-century CO₂ vegetation forcing on heat waves relative to end-of-21st-century CO₂ radiative forcing. We calculate the impact of end-of-21st-century CO₂ vegetation forcing on climate by subtracting the final 30 years of data in RadCO2 (end-of-21st-century radiative CO₂ forcing) from the final 30 years of data in TotalCO2 (end-of-21st-century total CO₂ forcing, see Table 2). Likewise, we calculate the impact of end-of-21st-century CO₂ radiative forcing on climate by subtracting the final 30 years of data in VegCO2 (end-of-21st-century vegetation CO₂ forcing) from the final 30 years of data in TotalCO2 (end-of-21st-century total CO₂ forcing, see Table 2). We choose to analyze the final 30 years of each simulation (average CO₂ concentrations ~ 984 ppm) in order to assess modeled CO₂ vegetation forcing and CO₂ radiative forcing that are generally consistent with a high emissions scenario projection (RCP8.5) for the end of the 21st century (58).

We emphasize that we use the RadCO2 simulations to back out the influence of CO₂ vegetation forcing on climate (i.e., TotalCO2 – RadCO2) rather than using the VegCO2 simulations directly (and vice versa for CO₂ radiative forcing) because it allows for an assessment of CO₂ vegetation forcing (CO₂ radiative forcing) relative to future CO₂ radiative forcing (CO₂ vegetation forcing). – Page 21-22, Lines 489-505

Second, to assess the influence of different levels of CO₂-forcing on our results, we also analyze the vegetation-driven responses of extreme heat metrics for CO₂ concentrations consistent with the middle 21st century in a high emissions scenario (~575 ppm). To assess the impact of mid-century CO₂ vegetation forcing we subtract the 30-year time period between years 58 and 87 of the RadCO2 simulation from the corresponding 30-year time period in the TotalCO2 simulation (Table 2). Simulation years 58 – 87 are chosen to reflect CO₂ values that are roughly consistent with years 2040 – 2070 in the RCP8.5 pathway (58). – Page 24, Lines 555-562

- Calculation of LAI of course also depends on assumptions about turnover as well as the GPP fixed and the carbon allocation. So there are further reasons for models differences not given in the text. Turnover should be added to this sentence.

Thank you for pointing this out. We now include turnover in the list of processes that influence the simulation of LAI.

The text now states:

Calculations of LAI are prognostic in all ESMs and depend on plant carbon stocks (in part determined by CO₂ fertilization), carbon allocation (the distribution of carbon to leaves, roots, and stems), and leaf turnover rates, as well as climate factors including temperature, soil moisture, and sunlight (37).– Page 25, Lines 575-578

Results:

- Given the intended publication is nature communications, it doesn't make sense to me to start with a discussion about LAI as the starting point of the results? Surely that authors would start with their key result and work backwards from there? Just a thought. In fact, is the whole chunk of text about model skill at simulating LAI strictly necessary given the focus of the paper? At best, this is supplementary material? It doesn't really say anything relevant for a paper on heatwaves.

As suggested, we have moved the discussion about the model skill at simulating LAI to the Supplementary Information.

The text now states:

An evaluation of LAI in each model is provided in the Supplementary Information (Supplementary Note 1 and Supplementary Fig. S1) and in (37).– Page 6, Lines 153-155

While we appreciate the author's very reasonable suggestion to highlight the heat wave result first, we have decided to maintain the figure order from the original manuscript. It is our intention to first show the reader that in most vegetated regions, CMIP5 models project the physiological impacts of CO₂ on surface water fluxes to outweigh the fertilization impacts of CO₂ on surface water fluxes. This in itself

is an interesting result, and one that has been debated in the modeling literature (e.g. Kergoat et al. 2002). From there, the figures tell a logical, step-by-step story in which we show that the reductions in transpiration lead to decreases in evaporative fraction and cloud cover/rainfall, which then contributes to warmer temperatures and more heat waves. We prefer this easy-to-follow narrative style.

Reference:

Kergoat, L., S., Lafont, H. Douville, B. Berthelot, G. Dedieu, S. Planto, and J.-F. Royer, 2002: Impact of doubled CO₂ on global-scale leaf area index and evapotranspiration: Conflicting stomatal conductance and LAI responses. *J. Geophys Res.*, 107, 4808.

- Fig 1a/b: I think simply the average LAI change is meaningless? Would a zonal plot be more instructive?

We have replaced the two panels (old Figure 1a-b) that showed the global mean change in LAI and transpiration versus CO₂ with six new panels (new Figure 1a-f) that display the zonal means of LAI and transpiration change versus CO₂ for (a-b) the Tropics (15°S – 15°N), (c-d) the Subtropics (15°S/N – 30°S/N), and (e-f) the Extratropics (30°S/N – 70°S/N).

We have also added zonal mean plots of the change in LAI and the change in transpiration (final 30 years of the simulations) for each model to the right side of the map plots (new Figure 1g-h).

We have updated the values in Supplementary Table 2 to reflect the tropical/subtropical/extratropical zonal averages (rather than the global averages) of the mean rate of LAI change ($\Delta\text{LAI}/\Delta\text{CO}_2$ ppm) and transpiration change ($\Delta\text{Tran}/\Delta\text{CO}_2$ ppm) during the first and second doubling of CO₂.

- Fig 1c/d: I find it interesting that none of this suite of models seemingly predict that LAI changes cancel any stomatal reduction in *g_s* (expressed as transpiration) anywhere on the globe? Is that just true for the regions in which models agree? Or is this true for each of the models simulations? I find this a very surprising result and would certainly think it warrants some further investigation.

Transpiration decreases in most regions despite increases in LAI. As you note, there are no regions where the models agree on a statistically significant increase in transpiration (Figure 1h). However, within each individual model, there are regions where increases in LAI lead to greater transpiration. These regions can be seen in Supplemental Figure 2s-x and Supplemental Figure 3a-f. Increases in transpiration are found primarily in water-limited regions which exhibit the greatest percent increase in LAI from elevated CO₂.

We discuss this interesting result in the text:

Increases in summer season transpiration due to CO₂ vegetation forcing are primarily confined to grassland and steppe regions in semi-arid portions of North America, Asia, Africa, and Australia, covering between 9.4% (CESM1-BGC) and 37.62% (BCC-CSM1-1) of total land area in the models (Fig. 2a and Supplementary Figure 2s-x). These regions of increasing summer transpiration exhibit relatively low summer LAI in the reference climate simulations (Supplementary Fig. 2a-f) coupled with large percentage increases in LAI in response to CO₂ vegetation forcing (Supplementary Fig. 3a-f). - Pages 7-8, Lines 179-186

- I have to confess that I got very little out of Fig 2. I either didn't follow it, or it needs a bit of work to make it clearer. Fig 2b in particular?

The objective of Figure 2a is to show that most vegetated grid cells exhibit a simultaneous increase in LAI and decrease in transpiration. The primary exception to this rule is in semi-arid regions where CO₂ fertilization has a very large impact and transpiration actually increases. We have altered the text describing these figures slightly to make this point more clear.

We have modified the text to state:

Increases in summer season transpiration due to CO₂ vegetation forcing are primarily confined to grassland and steppe regions in semi-arid portions of North America, Asia, Africa, and Australia, covering between 9.4% (CESM1-BGC) and 37.62% (BCC-CSM1-1) of total land area in the models (Fig. 2a and Supplementary Figure 2s-x). These regions of increasing summer transpiration exhibit relatively low summer LAI in the reference climate simulations (Supplementary Fig. 2a-f) coupled with large percentage increases in LAI in response to CO₂ vegetation forcing (Supplementary Fig. 3a-f). This reveals that, in general, unless LAI increases substantially, a vegetated grid cell will exhibit reduced transpiration in response to future CO₂ vegetation forcing from high CO₂. – Pages 7-8, Lines 179-188

We have removed the previous Fig. 2b because it did not add much new information. In its place, we now use the previous Supplementary Figure 3b. The new Fig. 2b show the global average percent of transpiration to total evapotranspiration (T/ET) for each model and the observations. It also shows the global average change in transpiration for each model. We moved this former Supplementary Figure to the Main Figures because it helps the reader clearly see that models that underestimate T/ET show small changes in transpiration in response to CO₂ vegetation forcing.

- Please quantify this: "the reductions in transpiration from CO₂ vegetation forcing in the models are large enough". How big are the changes (? mm/summer)?

We have quantified the global mean summer transpiration changes (mm/summer) and now include the range of values from the models in the text.

Specifically, the text now states:

Though most earth system models underestimate the global ratio of transpiration to total ET (Fig. 2b) (39), the reductions in transpiration from CO₂ vegetation forcing in the models, which, globally averaged range from -7.5 mm summer⁻¹ in BCC-CSM1-1 to -29.5 mm summer⁻¹ in CESM1-BGC, are large enough to drive statistically significant summer ET declines in most forested regions (Supplementary Fig. 4a-f). –Page 8, Lines 195-199

Discussion:

- I have no idea what this means: "Several models from CMIP5 do not include the plant physiological effects of CO₂", please clarify.

In several CMIP5 models (none of which were analyzed in this study), stomatal conductance does not depend on the atmospheric CO₂ concentration. Therefore, changes in CO₂ concentration do not directly cause changes in stomatal conductance. A breakdown of which models do and do not include these direct CO₂ physiological effects is provided by DeAngelis et al. (2016).

To make this point more clear, we have expanded the text to state:

It is also important to note that several models from the larger CMIP5 ensemble (none of which are analyzed in this study) do not include the physiological effects of CO₂, as stomatal conductance in those models does not depend on CO₂ (45). Our results suggest that projections from those models would underestimate future changes in extreme heat. Ensuring that all models include the dependence of stomatal conductance on CO₂ concentration may help to constrain future changes in heat extremes. –Page 18 , Lines 420-426

Reviewer 2 Comments

This paper quantifies how the response of plants to rising CO₂ (both through leaf area and physiological changes) influences heat wave occurrence, length, and intensity. They find that

heat waves are increased by plant responses, particularly in places with very high leaf area (lai) to begin with, i.e. the tropics and mid and high latitudes with high lai. Even with increases in lai under high CO₂, the decreases in water flux due to stomatal closure are large enough to lead to higher maximum temperatures. These results disagree with prior work showing that in Europe, a 2003-type heat wave is alleviated by plant responses when they save water from earlier in the season due to reduced stomatal conductance allowing for higher transpiration during the peak of the heat wave and thus reductions in peak temperatures.

This analysis is a novel and useful contribution to our understanding of how the coupled Earth system responds to change. It helps to illustrate that plants play a role in many climate impacts, even if those influences are not always identified in coupled simulations and often de-facto attributed to the radiative effects of CO₂.

I think the paper is appropriate for publication in Nature Communications. I have two major comments. I also have a number of very minor comments.

Thank you for this positive evaluation.

Major comments:

1. This paper doesn't cite Sellers et al. 1996!

Sellers et al. 1996 establishes the framework and concept that this work is based on by showing that temperature is influenced by the physiological response of plants to rising CO₂. The authors cite many papers that are derived following Sellers, but not the original. The authors can obviously fix this, but it is an egregious error! It seems like this may be needed elsewhere in the introduction, such as the first paragraph, but at a minimum in the following two locations:

- i. "due to the non-radiative effects of higher CO₂ is known as CO₂ physiological forcing (17)" Sellers et al 1996 is the appropriate reference here. Betts follows on Sellers.
- ii. "Meanwhile, CO₂ physiological forcing limits transpiration and enhances the ratio of sensible to latent heat fluxes at the leaf surface, increasing boundary layer temperatures (22)." Also here, covered by Sellers far earlier than Cao.

Thank you for pointing out our oversight. We agree that Sellers et al. 1996 must be cited in this paper. We have included references to Sellers et al. 1996 in both of the suggested locations.

Specifically, the text now states:

The closing of stomata and subsequent reduction in stomatal conductance and transpiration (evaporation of water from the leaf interior) due to the non-radiative effects of higher CO₂ is known as CO₂ physiological forcing (17, 18).– Pages 3-4, Lines 86-89

Meanwhile, in regions that are not severely water-limited, CO₂ physiological forcing limits transpiration and enhances the ratio of sensible to latent heat fluxes at the leaf surface, increasing boundary layer temperatures (17, 26, 27). – Page 4, Lines 97-100

Sellers et al. (1996) is now reference 17 in the manuscript.

2. There is no formal comparison of radiative effects on heat waves vs. vegetative effects on heat waves relative to the combined effect.

The relative sizes of radiative vs. vegetative effects was mentioned in the abstract (veg is 30-50% of radiative effect), but then not formally quantified elsewhere in the text. The parallel plots for radiative effects are shown in figure S6 and S7, but I would like to see something that quantifies the relative size of the two terms, and how they contribute to the total change in heat waves. This seems like a critical piece of information in assessing if the vegetative influence matters or not. The authors should have all information necessary to make this assessment. I feel that it is needed for the manuscript to tell a complete story. I realize that the radiative effects are larger, but it is important to know the relative contribution of the vegetation effects to the total response.

Thank you for this suggestion. We now include plots depicting the ratio of CO₂ vegetation forcing to CO₂ radiative forcing for two heat wave metrics: total heat wave days (HWTD) and heat wave maximum intensity (HWMI) (Figure 4e-f). We also include the change in all four heat wave metrics from total CO₂ forcing (derived from subtracting the first 30 years of the TotalCO₂ simulation from the final 30 years of the TotalCO₂ simulation, using the first 30 years of TotalCO₂ as the baseline period to define the temperature thresholds and heat waves) in the zonal average plots of Figure 4a-d.

Though we agree it would be useful to know what fraction of the total projected heat wave changes for the end of the 21st century can be attributed to the individual CO₂ vegetation and CO₂ radiative forcings, this is unfortunately not possible given the experimental design of the CMIP5 simulations. There are a few reasons why this individual contribution calculation is not possible: (1) The baseline periods used for the calculations of heat waves for the total CO₂ forcing, the CO₂ vegetation forcing, and the CO₂ radiative forcing are different (meaning, the changes are not relative to the same baseline heat waves). (2) Even if the baseline periods were the same, heat waves are discrete events, and quantifying the individual vegetation and radiative contributions to total CO₂-driven heat wave changes is therefore not possible. As an example, the end of 21st century CO₂ radiative forcing may be large enough to result in 100 more heat wave days during the summer (out of 120 days) (see for

example, Figure 6g-1). The CO₂ vegetation forcing from the same level of CO₂ concentrations may be enough to result in 40 more heat wave days (see for example, Figure 6a-f). Together, CO₂ vegetation forcing and CO₂ radiative forcing would account for an increase in 140 summer heat wave days out of 120 total heat wave days.

Your comment was very constructive however because it prompted us to include a quantification of the CO₂ vegetation forcing and CO₂ radiative forcing impacts on heat waves within the context of the historical period reference climate shown in Fig. S8. We now include a quantification of the CO₂ vegetation forcing on heat waves by subtracting the first 30 years of VegCO₂ from the final 30 years of VegCO₂ (in this case, we define the temperature thresholds and baseline period heat wave metrics from the first 30 years of VegCO₂). This can be compared with the impact of CO₂ vegetation forcing derived from subtracting the final 30 years of RadCO₂ from the final 30 years of TotalCO₂ (using the final 30 years of RadCO₂ as the baseline period for heat wave metrics). Likewise we include a quantification of the CO₂ radiative forcing on heat waves by subtracting the first 30 years of RadCO₂ from the final 30 years of RadCO₂ (in this case, we define the temperature thresholds and baseline period heat wave metrics from the first 30 years of RadCO₂). This can be compared with the impact of CO₂ radiative forcing derived from subtracting the final 30 years of VegCO₂ from the final 30 years of TotalCO₂ (using the final 30 years of VegCO₂ as the baseline period for heat wave metrics). We find that the choice of baseline period does not substantially influence the results. These additional heat wave analyses indicate that the changes in heat waves from the individual CO₂ forcings are fairly robust to background climate state, whether it is the final 30 years of RadCO₂ or a lower-forcing benchmark within each experiment.

We have added text detailing the new methods for quantifying the CO₂ vegetation and CO₂ radiative impacts on heat waves:

We provide two sensitivity analyses for our results. First, to assess the influence of our chosen reference climate on changes in heat extremes, we also analyze the change in heat wave indices within (rather than across) model simulations by subtracting the first 30 years of data in the VegCO₂ (RadCO₂) simulation from the final 30 years of data in the VegCO₂ (RadCO₂) simulation (Table 2). We also quantify the total CO₂-driven (CO₂ vegetation forcing + CO₂ radiative forcing) response of heat extremes by subtracting the first 30 years of the TotalCO₂ simulation from the final 30 years of the TotalCO₂ simulation. Accordingly, in these analyses, the reference period temperature thresholds and heat waves are defined using the first 30 years of data in VegCO₂, RadCO₂, and TotalCO₂. -

Pages 23-24, Lines 546-554

We have added new text describing the results from the new methods for quantifying the CO₂ vegetation and CO₂ radiative impacts on heat waves:

The simulated changes in heat wave metrics from CO₂ vegetation and CO₂ radiative forcing are robust to the choice of the reference climate (see Methods). The impacts of elevated end-of-21st-

century CO₂ vegetation forcing and CO₂ radiative forcing within a historical period reference climate (relatively low CO₂) are very similar to those within a future reference climate (relatively high CO₂, see Supplementary Fig. S8). However, small differences in the changes in heat wave metrics from CO₂ vegetation forcing are present at high latitudes, where the choice of reference climate (low versus high CO₂) likely has a large impact on summer vegetation growth (compare Fig. 4a-d with Supplementary Fig. S8a-d). – **Page 12, Lines 278-286**

Minor Comments:

It would be very helpful if the authors provided line numbers, or at a bare minimum page numbers. As such, I have tried to give approximate locations for my comments and include some text for context. I assigned the page numbers starting with the first page of the document as page 1.

We apologize for this frustrating oversight and we appreciate your efforts to provide clear and organized feedback in the absence of line numbers. We have added page numbers and line numbers to the revised manuscript.

- The names of the six ESMs and references for each model need to be in the paper somewhere, either inline in the text or in a table. I expected to find them in Table 1 but was surprised that was only describing the experiments. I see the names of the models are in the supplemental table, but still not the references. I think the references should be in the main reference list.

We now include the names of the six ESMs in the Methods section of the main text. The references for the six models are now included in the References section.

Specifically, we have modified the text to state:

We analyze simulations from six ESMs archived as part of the “carbon-climate feedback experiment” within CMIP5 (Table 1, Supplementary Table 1) (51). The six ESMs are BCC-CSM1-1 (52), CanESM2 (53), CESM1-BGC (54), IPSL-CM5A-LR (55), HadGEM2-ES (56), and MPI-ESM-LR (57), and are chosen based on the availability of daily-scale temperature data needed for the analysis of heat extremes. – **Page 20, Lines 467-471**

- (beginning of results section) These simulations do not include land use, so we shouldn't expect them to exactly capture observed LAI even in the present day. Perhaps this should be mentioned.

This is a good point. We have modified the text to state:

*Some of the regional-scale LAI differences between AVH15C1 and the models may be driven by a lack of prescribed land use change in several of the ESMs. A detailed evaluation of simulated LAI in the CMIP5 ensemble can be found in (1). **Supplementary Note 1***

As suggested by Reviewer 1, we have moved the discussion of model skill in simulating LAI to the Supplementary Information.

- (end of page 10, beginning of page 11) "positive" cloud cover is confusing. Could the authors use "increasing" instead?

Agreed. We have changed the two sentences to say:

*In general, the greatest summer near-surface warming (daily maximum temperature) occurs in the mid and high latitudes where the evaporative fraction, cloud cover, and rainfall decrease (Fig. 3a, and Supplementary Figs. 4g-x and 5a-f). Increases in mean daily maximum tropical temperatures are slightly smaller despite large reductions in transpiration because cloud cover and rainfall change very little or even increase in some areas in response to CO₂ vegetation forcing (Supplementary Figs. 4m-x and 5a-f) (35). – **Page 9, Lines 206-212***

- (top of page 12) "In comparison to the pattern of heat wave day changes from CO₂ radiative forcing, CO₂ vegetation forcing primarily impacts wetter, vegetated areas of the mid and high latitudes (Figs. 4A and S6A-L)." It would be helpful to move the figure reference (S6A-L) to immediately follow this first statement since that is where that quantity is shown.

Thank you for pointing this out. We have changed the sentence to say:

*In comparison to the pattern of heat wave day changes from CO₂ radiative forcing (see Methods, and Supplementary Fig. 6a-l), CO₂ vegetation forcing primarily impacts wetter, vegetated areas of the mid and high latitudes (Fig. 4a). – **Page 10, Lines 241-244***

- (middle of page 14) "slightly greater total-column soil moisture at the start of the summer season in most models (see Table S1 for the hydrologically active soil column depths in each model)" How did the authors define "hydrologically active" soil column depth that differs from total soil column depth? CESM1-BGC, for instance, has a depth of 5m, but is listed in the table as 2.86m for hydrologically active. I can't find an explanation of how this was determined and I don't find it obvious.

We determined which layers were hydrologically active through a combination of (1) reading model documentation and (2) analyzing model output variables that included soil depths (for example, we looked at the “mrlsl” variable, which lists the moisture content in each soil layer).

It turns out we were listing the center depth of the lowest hydrologically active soil layer, rather than the bottom depth of that layer. We have adjusted the depths in Supplementary Table 1 to reflect the depth of the bottom of the lowest soil layer. For example, the center depth of the lowest hydrologically active CESM layer is 2.86m, but the bottom of that layer actually reaches to 3.8m.

The new soil depths are:

BCC-CSM1-1 = 3.43m

CanESM2 = 4.1m

CESM1-BGC = 3.8m

HadGEM2-ES = 3.0m

IPSL-CM5A-LR = spatially variable (no change from previous manuscript)

MPI-ESM-LR = spatially variable (no change from previous manuscript)

In the case of CESM1-BGC, the documentation notes:

“The upper 10 layers are hydrologically active (i.e. the ‘soil’ layers) while the bottom five layers (3.8 m to 42 m depth) are thermal slabs that are not hydrologically active” (Lawrence et al., 2011).

Reference:

Lawrence DM, *et al.* (2011) Parameterization improvements and functional and structural advances in Version 4 of the Community Land Model. *Journal of Advances in Modeling Earth Systems* 3(1):n/a-n/a.

Reviewer 3 Comments

Overall Review

The manuscript addresses an important topic: how changes in vegetation LAI and stomatal physiology induced by elevated CO₂ alter land-atmosphere feedbacks and precisely heat waves. The authors analyze results from CMIP5 model ensemble and specifically simulations that include the role of CO₂ only on the radiative component of the models (radiative forcing) and only on vegetation component (physiological and fertilization forcing) (Table 1). In this way, they can separate the two effects. The main findings are that decreased stomatal conductance during summer generally reduces transpiration despite an increase in LAI (Fig. 2), which leads to an enhancement of future heat-waves frequency and intensity (Fig 4). As far as I know, only a couple of studies have looked at similar questions (Leomordant et al 2016; Kala et al 2016) but they had a more regional focus and do not analyze Earth System Model projections in such systematic way. Furthermore, the results of the current article are also different from results in Leomordant et al 2016. Overall, despite some concern on the plant-physiology implemented in CMIP5 Earth System Models, I think the study is well carried-out, well presented, and results are interesting and important. In summary, I enjoyed reading this research. However, in my opinion, there are a couple of aspects that need more discussion.

Thank you for this positive evaluation.

(i) One of the major concern I have is related to the stomatal conductance parameterization used in the various ESMs (Table S1). This is not something the authors can change but deserves a bit more attention in the discussion. Current ESMs use static parameters and employ a stomatal model that forcefully close stomata considerably in response to CO₂. This is a main driver of all the results presented in the article. Now, there is evidence that this is happening for a large number of species, but there is also evidence that it does not happen for some other species (e.g., Field et al 1995; Medlyn et al 1999; Keel et al 2007). In other words, the adopted stomatal parameterization is rather empirical and not flexible enough to accommodate for a range of real observed behaviors. This has been already discussed before in other articles (Damour et al, 2010 Paschalis et al 2017) but needs an explicit mention because it can affect most of the results. It is not just a parameterization problem (e.g., Kala et al 2016) as the authors also discusses in this paper but likely a structural model problem. Furthermore, the response of LAI to increase CO₂ is also quite difficult to simulate with current ESMs (see discussion in Fatichi et al 2016), which adds an additional level of uncertainty that needs to be explicitly mentioned.

Thank you for pointing to these references. We agree that the points you bring up should be discussed in our Summary and Discussion section. We have included new text that states:

More broadly, the substantial role of vegetation physiology in shaping future simulated hydrology and surface energy fluxes in ESMs highlights the need to develop mechanistic models of plant growth

and physiology and to increase observational efforts towards understanding vegetation's role in the hydrologic cycle. Presently, models use semi-empirical formulations of stomatal conductance that do not capture the full range of stomatal behavior across plants (e.g., 31). Similarly, models struggle to simulate observed relationships between elevated CO₂ and changes in LAI (27). Both of these factors limit confidence in projections of regional climate change, such as those presented in this study, and point to the need for increased process-based understanding and mechanistic models of stomatal conductance and carbon allocation in ESMs (46, 47).– Pages 18-19, Lines 427-436

(ii) The study of Leomordant et al 2016 is based on actually observed summer of 2003 in Europe and some of the differences may be also related to the persistence of the heatwave in that summer. In other words if the period interested by heat-waves is short, temperature may be amplified by stomatal closure, however if the period is long the saved water can be used later in the summer to decrease sensible heat and temperature in other heat-waves that can occur later on. If ESMs do not capture well the current length, correlation and intensity of heatwaves, these effects may not be so evident in the analyzed simulations. ESMs may have issues in reproducing the most persistent weather patterns and therefore the most-extreme heat-waves also for present climate conditions. I wonder if a comparison of observed and simulated heat-wave statistics has been published previously and can be referred to, or if such a comparison can be made here and will strengthen the relevance of this manuscript. Another way, could be to look if the physiological effect is becoming less important as the dry season progresses, especially in water-limited climates. As a matter of fact, the most clear physiological effect on heatwave statistics is on the humid tropics (Fig. 4), which are less likely water-limited.

This is a good point. Russo et al. (2014) show that the CMIP5 models underestimate the number of extreme heat waves (using a combined metric of length and intensity) compared with reanalysis data, particularly in the mid-latitude Northern Hemisphere. It is possible that the soil moisture savings mechanism may reduce temperatures in particularly long and severe heat wave events and that these events aren't present in the models.

We have added text that states:

It is also important to note that the regional climate modeling work in (12) analyzed a particularly long and severe heat wave event (the European summer 2003 heat wave). While it is clear that CO₂ vegetation forcing enhances the intensity and frequency of future heat wave events within CMIP5 models in general, it is possible that CO₂ vegetation forcing may result in greater surface latent cooling and reduced temperatures during one of these anomalously long heat waves. ESMs from CMIP5 tend to underestimate the frequency of the most severe heat waves (44), and therefore may not be suited to fully assess the impact of CO₂ vegetation forcing on all types of heat wave events. – Page 17, Lines 396-404

Additionally, we analyzed the changes in heat waves for each month during the summer, but we found no significant changes between the early part of the summer and the later part of the summer in the CMIP5 simulations. As an example, we have included the change in the number of heat wave days during each summer month from CO₂ vegetation forcing for the IPSL-CM5A-LR model below:

(iii) Most of the results are presented and discussed for the last 30 years of the analyzed simulations that have a CO₂ concentration of roughly 1000 ppm. It would be interesting to have in the article some number as a reference (maybe some Figure in the Supp. Information) also for CO₂ concentrations of 500-600 ppm that are what we expect in the near future conversely to a far future.

We now include analysis of the impact of mid-21st century CO₂ vegetation forcing on heat waves (years 58 – 87 of the simulations; average CO₂ values ~ 575 ppm). The new results are presented in Supplementary Figure 8i-1. As expected, the impacts of the lower CO₂ concentrations on heat waves are smaller. However, the spatial patterns of heat wave change are similar to the high CO₂-driven response (except for the heat wave max intensity, which does not show a robust signal across the models).

In addition to Supplementary Figure 8i-1, we have included new text in the Methods section that states:

Second, to assess the influence of different levels of CO₂-forcing on our results, we also analyze the vegetation-driven responses of extreme heat metrics for CO₂ concentrations consistent with the middle 21st century in a high emissions scenario (~575 ppm). To assess the impact of mid-century CO₂ vegetation forcing we subtract the 30-year time period between years 58 and 87 of the RadCO2 simulation from the corresponding 30-year time period in the TotalCO2 simulation (Table 2). Simulation years 58 – 87 are chosen to reflect CO₂ values that are roughly consistent with years 2040 – 2070 in the RCP8.5 pathway (58). – Page 24, Lines 555-562

We have also included new text in the Results section that states:

Not surprisingly, changes in heat wave metrics from projected mid-21st century CO₂ vegetation forcing (average CO₂ ~575 ppm, see Supplementary Fig. 8i-1) are smaller than those from the projected end of 21st century forcing (average CO₂ ~984 ppm, see Fig. 4a-d), though the spatial patterns of extreme heat change (where statistically significant) are similar. – Page 12, Lines 286-290

**Sincerely,
Simone Fatichi**

SPECIFIC COMMENTS

Page 3. Line 3. I am not sure the meaning of “human morbidity” is generally known.

We have decided to keep the word “morbidity” in the text because it is commonly used in the scientific literature when discussing the potential impacts of heat waves on humans, and we would like to use terminology that is consistent with this previous work. It is a particularly useful word in this context because its definition covers a wide range of illnesses incited and exacerbated by extreme heat.

Page 3. Line 19. I would suggest to use “reduced” rather than “narrowed”.

We have modified the text to state:

Under high CO₂, plant photosynthetic carbon fixation rates increase, while stomatal aperture is reduced or maintained (14). – **Page 3, Lines 82-84**

Page 4. Line 8-10. and Page 5 Line 6-10. These statements are strictly true only if there is enough water in the soil that plants are never water stressed. If this is not the case, CO₂ can temporarily reduce transpiration but the saved water will be used anyhow during the growing-season and the integrated transpiration and evapotranspiration will be similar in any CO₂ or LAI scenarios. In a severely water limited region evapotranspiration is not a function of CO₂ or LAI but of the amount of precipitation regardless of CO₂ levels (e.g., Fatichi et al 2016).

Thank you for pointing this out. We have changed the text to indicate that CO₂ physiological forcing only reduces transpiration when water is not severely limiting.

Specifically, we have modified the text to state:

Meanwhile, in regions that are not severely water-limited, CO₂ physiological forcing limits transpiration and enhances the ratio of sensible to latent heat fluxes at the leaf surface, increasing boundary layer temperatures (17 26, 27). – **Page 4, Lines 97-100**

Page 15. “maintaining the same transpiration” rather than “limiting transpiration reduction”.

We have modified the text to state:

The combination of excess soil moisture at the start of the dry season, greater LAI during the dry season, deep roots, and high evaporative demand are likely responsible for maintaining the same transpiration during this time (Fig. 5c) (e.g. 12). – **Page 14, Lines 331-333**

Page 18. Line 1. Leomordant et al. 2016 also look to a very peculiar heat-wave in western Europe -the summer of 2003 - with vegetation that experience water limitations, at least in their simulations. This can be an additional reason to explain the difference. In any case, for the first part of the summer results agree with the CMIP5-ESMs results.

We now mention in the text that the anomalous length and severity of the 2003 European heat wave analyzed in Lemordant et al. (2016) may contribute to the difference between the CMIP5 results and the regional climate modeling result.

Specifically, we have added new text that states:

It is also important to note that the regional climate modeling work in (12) analyzed a particularly long and severe heat wave event (the European summer 2003 heat wave). While it is clear that CO₂ vegetation forcing enhances the intensity and frequency of future heat wave events within CMIP5 models in general, it is possible that CO₂ vegetation forcing may result in greater surface latent cooling and reduced temperatures during one of these anomalously long heat waves. ESMs from CMIP5 tend to underestimate the frequency of the most severe heat waves (44), and therefore may not be suited to fully assess the impact of CO₂ vegetation forcing on all types of heat wave events. – Page 17, Lines 396-404

Caption of Figure 2. “Median grid point % change in LAI for locations with” is a bit awkward. If I understood correctly I would write “Median change in LAI [%] over the grid points with ...”

Thank you for this suggestion.

We have removed the former Figure 2b from the manuscript because it did not add much new information that couldn't already be gathered from the other Main and Supplementary Figures.

Caption of Figure 3. The evaporative fraction is defined as the ratio of latent heat flux to the sum of latent and sensible heat fluxes and not as sensible heat over latent heat, which is the Bowen Ratio. Please correct. I guess, in the figure, you show changes in either the real evaporative fraction or the inverse of the Bowen ratio.

Thank you for catching this typo in our Figure caption. We were correctly plotting evaporative fraction (the ratio of latent heat flux to the sum of latent and sensible heat fluxes), but we mislabeled this in the caption. We have corrected the caption on Figure 3.

The Figure 3 caption now states:

Mean evaporative fraction (ratio of latent heat fluxes to the sum of sensible and latent heat fluxes).

Figure S8. The small map of the world contains polygons only for two selected regions rather than four.

Thank you for catching this. We have added the polygons to the two tropical locations in the new Supplementary Figure 9 (former Figure S8).

REVIEWERS' COMMENTS:

Reviewer #2 (Remarks to the Author):

I think that the revised manuscript is much improved, especially by reducing the focus on the validation of LAI and the Lemordant et al. paper. I am satisfied with the modifications and the response to reviewer comments. I recommend acceptance for publication.

Reviewer #3 (Remarks to the Author):

In my previous assessment, I had a very positive evaluation of the importance of the results presented in this manuscript for increasing the knowledge of vegetation-atmosphere interactions and for spurring further research on this topic. The authors addressed satisfactorily all my comments on the previous version. I just have a few final minor suggestions below.

Sincerely,

Simone Fatichi

SPECIFIC COMMENTS

Page 2. Line 45. Why only “to the direct effects”? The contribution of plants to changes in heat extremes will depend on plant responses to direct and indirect effects of CO₂. E.g., if LAI or vegetation stress changes because of soil-moisture change, this will also affect the future heat waves. This is actually, what you consider once you subtract RadCO₂ from TotalCO₂.

Page 11. Line 249-251. I do not think the fact that changes in heat wave statistics of vegetation forcing and radiative forcing do not sum up is only related to the “discrete” nature of heat wave event. It is rather related to the fact that the two effects are not independent and therefore they are not additive.

Page 14. Line 339. I would suggest the authors, if possible, to introduce sub-sections in the “Summary and Discussion”. As it is now it is a quite long and continuous section.

Page 19. Line 447. I think reference (34), which is purely a modeling study, is not appropriate to support the fact that CMIP5 models do a reasonable job in simulating impacts of CO₂ fertilization. As a matter of fact from De Kauwe et al 2013 and Medlyn et al 2015, it emerges that there are still many issues in models to reproduce the range of responses to elevated CO₂.

References

De Kauwe, MG, Medlyn BE, Zaehle S, Walker AP, Dietze MC et al (2013). Forest water use and water use efficiency at elevated CO₂: a model-data intercomparison at two contrasting temperate forest FACE sites. *Glob Chang Biol* 19:1759–1779.

Medlyn, BE, Zaehle S, De Kauwe MG, Walker AP, Dietze MC, Hanson PJ, et al (2015) Using ecosystem experiments to improve vegetation models. *Nat Clim Change* 2015, 5:528–534.

Response to Reviewers Document for “Amplification of Heat Extremes by Plant CO₂ Physiological Forcing”, by Skinner, Poulsen, and Mankin.

We thank the editor and reviewers for their insightful comments, which have improved the manuscript.

Reviewer 2 Comments

I think that the revised manuscript is much improved, especially by reducing the focus on the validation of LAI and the Lemordant et al. paper. I am satisfied with the modifications and the response to reviewer comments. I recommend acceptance for publication.

Thank you for this positive evaluation.

Reviewer 3 Comments

In my previous assessment, I had a very positive evaluation of the importance of the results presented in this manuscript for increasing the knowledge of vegetation-atmosphere interactions and for spurring further research on this topic. The authors addressed satisfactorily all my comments on the previous version. I just have a few final minor suggestions below.

**Sincerely,
Simone Fatichi**

Thank you for this positive evaluation.

SPECIFIC COMMENTS

Page 2. Line 45. Why only “to the direct effects”? The contribution of plants to changes in heat extremes will depend on plant responses to direct and indirect effects of CO₂. E.g., if LAI or vegetation stress changes because of soil-moisture change, this will also affect the future heat waves. This is actually, what you consider once you subtract RadCO₂ from TotalCO₂.

We have changed the text to state:

The contribution of plants to changes in future heat extremes will depend on the responses of vegetation growth and physiology to the direct and indirect effects of elevated CO₂. – Page 2, Lines 44-46.

Page 11. Line 249-251. I do not think the fact that changes in heat wave statistics of vegetation forcing and radiative forcing do not sum up is only related to the “discrete” nature of heat wave event. It is rather related to the fact that the two effects are not independent and therefore they are not additive.

We have changed the text to state:

Given the choice of different reference climate for the heat wave definitions (see Methods), and because the radiative and vegetation forcings are not independent, the individual changes in heat wave characteristics from CO₂ vegetation forcing and CO₂ radiative forcing are not expected to sum to the total CO₂ response. Pages 10-11, Lines 248-252.

Page 14. Line 339. I would suggest the authors, if possible, to introduce sub-sections in the “Summary and Discussion”. As it is now it is a quite long and continuous section.

We thank the reviewer for this suggestion. Our understanding is that Nature Communications does not recommend subheadings in the Discussion Section.

Page 19. Line 447. I think reference (34), which is purely a modeling study, is not appropriate to support the fact that CMIP5 models do a reasonable job in simulating impacts of CO₂ fertilization. As a matter of fact from De Kauwe et al 2013 and Medlyn et al 2015, it emerges that there are still many issues in models to reproduce the range of responses to elevated CO₂.

We have removed the sentence that referenced the modeling study (34).